# Phosphoinositide-binding proteins mark, shape and functionally modulate highly-diverged endocytic compartments in the parasitic protist *Giardia lamblia*

**Lenka Cernikova[1], Carmen Faso[1,2]ᴑ\*, Adrian B. Hehl[1]ᴑ\***

**1** Institute of Parasitology, University of Zurich, Zurich, Switzerland, **2** Institute of Cell Biology, University of Bern, Bern, Switzerland

ᴑ These authors contributed equally to this work.
\* carmen.faso@izb.unibe.ch (CF); adrian.hehl@uzh.ch (AH)

**Data Availability Statement:** All relevant data are within the manuscript and its Supporting Information files.

## Abstract

Phosphorylated derivatives of phosphatidylinositol (PIPs) are key membrane lipid residues involved in clathrin-mediated endocytosis (CME). CME relies on PIP species $PI(4,5)P_2$ to mark endocytic sites at the plasma membrane (PM) associated to clathrin-coated vesicle (CCV) formation. The highly diverged parasitic protist *Giardia lamblia* presents disordered and static clathrin assemblies at PM invaginations, contacting specialized endocytic organelles called peripheral vacuoles (PVs). The role for clathrin assemblies in fluid phase uptake and their link to internal membranes via PIP-binding adaptors is unknown. Here we provide evidence for a robust link between clathrin assemblies and fluid-phase uptake in *G. lamblia* mediated by proteins carrying predicted PX, FYVE and NECAP1 PIP-binding modules. We show that chemical and genetic perturbation of PIP-residue binding and turnover elicits novel uptake and organelle-morphology phenotypes. A combination of co-immunoprecipitation and *in silico* analysis techniques expands the initial PIP-binding network with addition of new members. Our data indicate that, despite the partial conservation of lipid markers and protein cohorts known to play important roles in dynamic endocytic events in well-characterized model systems, the *Giardia* lineage presents a strikingly divergent clathrin-centered network. This includes several PIP-binding modules, often associated to domains of currently unknown function that shape and modulate fluid-phase uptake at PVs.

## Author summary

In well-characterized model eukaryotes, clathrin-mediated endocytosis is a key process for uptake of extracellular material and is regulated by more than 50 known proteins. A large number of these carry phosphoinositide (PIP)-binding domains and play a central role in the regulation of endocytosis. Here, we report on the detailed functional characterization of PIP-binding proteins in the intestinal parasitic protist *Giardia lamblia*. We show evidence that proteins carrying specific PIP-binding domains are directly involved

**Funding:** ABH was supported by Swiss National Science Foundation grants 140803 and 125389; CF was supported by Swiss National Science Foundation grant PR00P3_179813 (www.snsf.ch). The funders played no role in the study design, data collection and analysis, decision to publish, or preparation of the manuscript.

**Competing interests:** The authors have declared that no competing interests exist.

in fluid-phase uptake. Furthermore, using co-immunoprecipitation assays, we confirm these proteins' association to *G. lamblia's* clathrin assemblies. In addition, using state-of-the-art imaging strategies, we demonstrate a previously unappreciated level of complexity involving PIPs and their partner proteins in marking and shaping *G. lamblia*'s unique endocytic compartments. Our data contribute substantially to an updated working model for *G. lamblia*'s host-parasite interface, demonstrating how uptake in this parasite is directly regulated by a variety of PIP residues and PIP-binding modules, which have been re-routed from conserved pathways, likely as a result of host-parasite co-evolution.

## Introduction

Phosphorylated derivatives of the minor membrane phospholipid phosphatidylinositols (PIPs) are surface molecules of most eukaryotic endomembrane compartments [1–3]. PIPs play important roles in diverse pathways including signaling cascades, autophagy and membrane remodeling [2, 4–8]. Their diverse functions are reflected in their distinct subcellular distribution. $PI(4,5)P_2$ is highly enriched at the plasma membrane (PM) together with $PI(3,4,5)P_3$ [4, 5]. PtdIns(4)P's largest pool is at Golgi membranes, with lower density found at the PM. $PI(3)$ P is converted into $PI(3,5)P_2$ on early endosomes during transition to multivesicular bodies and then late endosomes [6, 7]. $PI(3)P$ is also a marker of phagosomes [8] while $PI(5)P$ marks both the PM and endomembranes [9]. At least 14 distinct PIP-binding modules have been identified in eukaryotes, demonstrating a wide range of selective protein-lipid interactions associated with the PM and internal membranes [10].

In addition to their structural functions in membranes, in model eukaryotes PIPs are involved in spatiotemporal organization of membrane remodeling processes such as clathrin-coated vesicle (CCV) formation during clathrin-mediated endocytosis (CME). In particular, $PI(4,5)P_2$ marks sites of endocytosis at the PM and recruits proteins involved in the formation of CCVs [11]. The protein interactomes of mammalian $PI(4,5)P_2$-binding proteins include the early-acting clathrin interacting partners AP2 [12–15], AP180/CALM [16, 17] and epsin [17, 18]. These factors carry specific PIP-binding domains that can discriminate between PIP variants to achieve membrane targeting specificity.

*Giardia lamblia* (syn. *intestinalis*, *duodenalis*) is a widespread parasitic protist that colonizes the upper small intestine of vertebrate hosts. Its life cycle is marked by the alternation of an environmentally-resistant, infectious cyst stage responsible for parasite transmission, and a trophozoite stage proliferating by binary fission. Nutrient uptake of trophozoites in the lumen of the small intestine is almost entirely routed through peripheral vacuoles (PVs). These organelles are positioned just beneath the PM and are contacted by funnel-shaped invaginations of the PM that are likely conduits for uptake of fluid-phase extracellular material [19].

A recent characterization of the PV protein interactome, using the conserved *G. lamblia* clathrin heavy chain (*Gl*CHC) as affinity handle, confirmed the endocytic nature of these organelles by highlighting the presence of giardial AP2 (*Gl*AP2) subunits, the single dynamin-like protein *Gl*DRP and a putative clathrin light chain *Gl*4259 (*Gl*CLC; [19]). Notably absent were components for CCV uncoating and disassembly, consistent with a lack of measurable clathrin assembly turnover and in line with observations that CCVs are missing in *G. lamblia* and clathrin assemblies are static and long-lived. Therefore, *G. lamblia* presents an unusual endocytic system, characterized by divergent endocytic compartments (PVs) associated to static clathrin assemblies that are not predicted to form ordered arrays or higher-order structures such as CCVs, yet are closely membrane-associated.

Included in the giardial CHC interactome were three proteins with predicted PIP-binding domains: FYVE domain protein *Gl*16653 and two PX-domain proteins (*Gl*7723 and *Gl*16595), the latter part of a six-member protein family (Table 1; [19, 20]). In a previous study, we hypothesized that *Gl*16653 (*Gl*FYVE), *Gl*7723 (*Gl*PXD1) and *Gl*16595 (*Gl*PXD2) act as PIP-binding adaptors to link and maintain static clathrin assemblies at the PM and PV membrane interface in *G. lamblia* [19]. We further postulated that a perturbation of PIP-binding protein levels and/or function would lead to impaired fluid-phase uptake by affecting PV functionality. To test these hypotheses, we performed an in-depth functional characterization of all previously-identified PIP-binding proteins associated to clathrin at PVs. We assessed their lipid-binding preferences and visualized their subcellular localizations using electron microscopy and both conventional and super resolution light microscopy. By manipulating protein levels and/or function we could elicit novel fluid-phase uptake and PV morphology-related phenotypes, thereby establishing PIPs as a link between the role of clathrin as a membrane remodeling protein and PV-based endocytosis in *G. lamblia*. Furthermore, we used a combination of co-immunoprecipitation and *in silico* annotation techniques to expand protein interactomes established previously, thereby discovering a new set of PIP-binding proteins with roles likely reaching beyond the PV compartment. Lastly, we propose an updated working model summarizing the complex networks between PIP-binding proteins and clathrin assemblies at PVs.

## Results

### The *G. lamblia* genome encodes at least seven distinct PIP-binding modules

Given that several types of PIP-binding modules have been identified in eukaryotes, we determined how many endocytosis-associated module types were actually represented in the *Giardia* genome, in addition to the known *G. lamblia* epsin, FYVE and PXD variants [19–23]. For this reason, we selected a total of 14 protein types from various organisms known to harbour PIP-binding domains, some of them involved in endocytosis. These are: ANTH (AP180 N-terminal homology), ENTH (epsin N-terminal homology), PH (Pleckstrin homology domain), FYVE (Fab1, YOTB, Vac1 and EEA1), PX (Phox homology), BAR (bin, amphiphysin and Rvs), FERM (4.1, ezrin, radixin, moiesin), PROPPINs (β-propellers that bind PIs), C2 (conserved region-2 of protein kinase C), GOLPH3 (Golgi phosphoprotein 3), PDZ (postsynaptic density 95, disk large, zonula occludens), PTB (phosphotyrosine binding), Tubby modules and the PH-like module of the endocytosis-associated NECAP1 protein [24]. Representatives for each module were used as baits for the HMM-based tool HHpred [25] for protein structure prediction and the detection of remotely related sequences in the *G. lamblia* predicted proteome (Table 1). Putative *Giardia* protein homologs (Table 1) were then subjected to the online tools SMART [26, 27] and InterProScan [28] to identify conserved structural domains and sequence motifs within a query sequence (Fig 1A).

This data mining approach detected high-confidence homologs for hitherto undiscovered *G. lamblia* proteins containing PH-like, FERM, BAR, FYVE and PROPPINs PIP-binding domains (Table 1, Fig 1A). No homologs could be found for the ANTH, GOLPH3, PDZ, PTB, Tubby and PH PIP-binding module types. Specifically for the C2 PIP-binding module, we could not detect significantly-related *Giardia* sequences using the query sequence 1CZS [29] although ORF numbers GL50803_16728, GL50803_114201, GL50803_17406 and GL50803_14855 all code for predicted C2-containing phosphoinositide kinases [30, 31].

Protein GL50803_17195 (*Gl*NECAP1) is a predicted NECAP1 homolog containing a PH-like domain. Similarly, a conserved PH-like domain found at the C-terminus of FERM proteins was correlated with high confidence to protein GL50803_115468 (*Gl*FERM).

**Table 1. *G. lamblia* PIP-binding proteins.** A compilation of all PIP-binding domains identified in the Giardia Genome Database (www.giardiadb.org; GDB) using previously characterized domains [24] as baits for HMM-based homology searches (column 1). Predicted giardial orthologs are present for PIP-binding domains ENTH, PH, FYVE, PX, BAR, FERM and PROPPINs (column 2) and mostly retrieve the correct domains when used as baits for reverse HHpred searches (column 4). Except for *Gl*epsin, *Gl*PXD2 and *Gl*PROP1 and 2, all others are currently annotated on GDB as generically "hypothetical", *i.e.* of unknown function (column 6). Each orthologue was assigned a name used throughout this report (column 7). Functional domain predictions using SMART (http://smart.embl-heidelberg.de/; column 8) and subcellular localization data (column 9) either previously reported or acquired in this study (column 10), are also included.

| Domain (UniProtKB entry) | *G. lamblia* orthologs (UniProtKB entry, GDB gene_ID, Probability/ E-value | Seq. id./sim. (%/%)[1] | 3D structure homologue (HHPRED \*.pdb, Probability/E-value, Seq. id./sim. (%/%)[1], name used by HHPRED | 3D structure homologue (I-TASSER \*.pdb)[2] | *G. lamblia* ortholog annotation on GDB | Name used in this study | Domains (online tool SMART) | Localisation | References |
|---|---|---|---|---|---|---|---|---|---|
| ENTH (1h0a/ O88339) | A8BIK9 GL50803_3256 100/6e-37 | 28/56.1 | 6enr 100/2.4e-32 23/ 35 Epsin-1 | 3onk | EH domain binding protein epsin 2 | *Gl*epsin | Pfam: ENTH (4.8e-33) | Ventral disc/ PVs | Ebneter 2014, Touz 2015 |
| PH (1tqz/ Q9CR95) | A8BII4 GL50803_17195 (100/3.6e-55) | 36/61.3 | 1tqz 100/4.4e-52 36/ 61.3 NECAP-1 | 1tqz | Hypothetical protein | *Gl*NECAP1 | Pfam: DUF1681 (5.5e-43) | PVs/ Cytosolic | Zumthor 2016 |
| FYVE (1hyi/ Q15075) | A8BDZ8 GL50803_16653 (98.17/2.1e-9) | 39/92.1 | 3mpx 92.07/0.11 19/ 33.4 FYVE/RhoGEF and PH-domain containing protein | 1joc | Hypothetical protein | *Gl*FYVE | SMART: FYVE (6.38e-18) | PVs | Sinha 2011, Zumthor 2016 |
| | A8B288 GL50803_16801 (89.99/0.031) | 28/75.6 | 3t7l 99.08/3.4e-13 34/ 96.2 Zinc finger FYVE domain-containing protein | 1vfy | Hypothetical protein | *Gl*16801 | FYVE/PHD zinc finger (1.6e-05) | PVs | This study |
| PX (1h6h/ Q15080) | A8B343 GL50803_7723 (96.41/0.00011) | 20/36.4 | 3lui 97.19/2.5e-5 20/ 32.1 SNX-17 | 4oxw | Hypothetical protein | *Gl*PXD1 | SMART: PX (7.2e-11) | PVs | Zumthor 2016, Jana 2017 |
| | A8B344 GL50803_16595 (99.43/3.8e-15) | 18/21.1 | 10cs 97.64/1.0e-6 23/ 36.2 SNX GRD19 | 4on3 | Liver stage antigen-like protein | *Gl*PXD2 | SMART: PX (6.01e-11) | PVs | Zumthor 2016, Jana 2017 |
| | A8B341 GL50803_16596 (99.55/1.7e-16) | 18/23 | 4ikb 98.81/2.4e-10 22/ 38 SNX-11 | 3iq2 | Hypothetical protein | *Gl*PXD3 | SMART: PX (2.11e-06) | PVs | Jana 2017 |
| | A8B322 GL50803_42357 (99.51/6e-16) | 21/27.9 | 4pqo 98.18/2.9e-7 14/ 20.2 SNX-14 | 3iq2 | Hypothetical protein | *Gl*PXD4 | SMART: PX (1.62e-02) | PVs | Jana 2017 |
| | A8BIZ8 GL50803_16548 (99.56/1.6–16) | 17/16.3 | 4ikb 99.59/2.0e-16 20/ 32.5 SNX-11 | 2mxc | Hypothetical protein | *Gl*PXD5 | SMART: PX (2.03e-08) | PVs | Jana 2017 |
| | D3KH98 GL50803_24488 (-/-) | -3 | 4ikd 99.67/5.0e-18 15/ 35.5 SNX-11 | 1xtn | Hypothetical protein | *Gl*PXD6 | SMART: PX (1.89e-02) | PVs | Jana 2017 |
| BAR (2efl/q96ru3) | A8BZ00 GL50803_15487 (95.09/0.055) | 11/2.8 | 2v0o 96.41/0.047 11/ 14.7 FCH domain | 2v0o | Hypothetical protein | *Gl*BAR1 | - | PVs/ Cytosolic | Morrison 2007 |
| | A8BMB7 GL50803_14045 (95.67/0.027) | 10/4.5 | 1uru 95.53/0.25 16/ 20.6 Amphiphysin | 1uru | Hypothetical protein | *Gl*BAR2 | - | PVs/ Cytosolic | Morrison 2007 |

(*Continued*)

**Table 1.** (*Continued*)

| Domain | *G. lamblia* orthologs (UniProtKB entry, GDB gene_ID, Probability/ | Seq. | 3D structure homologue (HHPRED *.pdb, Probability/E-value, Seq. id./sim. (%/%)[1], name used by HHPRED | 3D structure homologue (I-TASSER *. pdb)[2] | *G. lamblia* ortholog annotation on GDB | Name used in this study | Domains (online tool SMART) | Localisation | References |
|---|---|---|---|---|---|---|---|---|---|
| (UniProtKB entry) | E-value | id./sim. (%/%)[1] | | | | | | | |
| FERM (1e5w/ P26038) | A8BC43 | 12/11.1 | 6d2k 100/2.9e-38 10/ 7.7 FERM,ERHGEF and PH-containing protein | 5mv9 | Hypothetical protein | *Gl*FERM | - | Cytosolic | This study |
| | GL50803_115468 | | | | | | | | |
| | (100/7.5e-37) | | | | | | | | |
| PROPPINs (4exv/ q6cn23) | A8B6Z3 | 21/36.3 | 3vu4 100/1.2e-36 23/ 38.1 KmHsv2 | 4exv | WD-40 repeat family protein | *Gl*PROP1 | SMART: WD40/ WD40 (9.6e- 02/1.79e-1) | PVs/ Cytosolic | Rout 2016 |
| | GL50803_10822 | | | | | | | | |
| | (100/7.5e-37) | | | | | | | | |
| | A8BTE2 | 16/25.9 | 5nnz 100/1.1e-36 14/ 8.5 | 4exv | WD-40 repeat family protein | *Gl*PROP2 | SMART: WD40/ WD40 (2.80e-03/ 14.8) | PVs/ Cytosolic | This study |
| | GL50803_16957 | | | | | | | | |
| | (100/7.5e-37) | | | | | | | | |

[1]Values for identity and similarity refer only to the predicted PIP-binding module. Seq. -Sequence. Id.—Identity. Sim.—Similarity.

[2]The .pdb identifiers in this column were used as modelling templates for *Giardia* PIP-binding proteins (S1 Fig).

[3]Protein GL50803_24488 was found by searching GDB for PXD protein paralogues.

Immunofluorescence assay (IFA) and confocal microscopy imaging of a tagged *Gl*FERM reporter detects a diffused cytosolic subcellular distribution (Fig 1B). In contrast, BAR domain-containing proteins GL50803_15847 and GL50803_14045 (*Gl*BAR1 and 2, respectively) localize in close proximity to PVs, similar to tagged reporters for PROPPINs GL50803_10822 (*Gl*PROP1) and GL50803_16957 (*Gl*PROP2) (Fig 1B).

Since *Gl*PXD1-2, *Gl*FYVE and *Gl*NECAP1 were experimentally shown to be associated to giardial clathrin assemblies [19], we selected these proteins and *Gl*PXD3-6 for more detailed subcellular localization experiments. Stimulated emission-depletion (STED) microscopy in co-labelling experiments with Dextran-OG as a marker for fluid-phase endocytosis unequivocally confirmed accumulation for *Gl*PXD1-4 and 6, *Gl*FYVE and *Gl*NECAP1 epitope-tagged reporters at PVs (Fig 1C–1I). The signal generated by *Gl*PXD5 reporters was insufficient for a conclusive localization using STED microscopy but was shown to localize in close proximity to PVs using conventional confocal microscopy (Fig 1B).

To extend the initial annotation of giardial PIP-binding proteins we performed multiple sequence alignment (MSA) analyses for each giardial PIP-binding module with selected orthologs to delineate lipid-binding motifs and residues critical for PIP recognition (S1 Fig). *In silico* structural analyses of the lipid-binding domains of giardial proteins and their closest homologs were performed *ab initio* using the online tool I-TASSER [32–34]. Comparative analysis of structure models generated with I-TASSER clearly demonstrated positional conservation of residues critical for PIP binding (S1 Fig).

Taken together, *in silico* analysis identifies seven distinct PIP-binding module types encoded in the *G. lamblia* genome, conserved on both sequence and structural levels. Subcellular localization of epitope-tagged variants by fluorescence microscopy indicates clear association to PVs with the exception of *Gl*epsin [21].

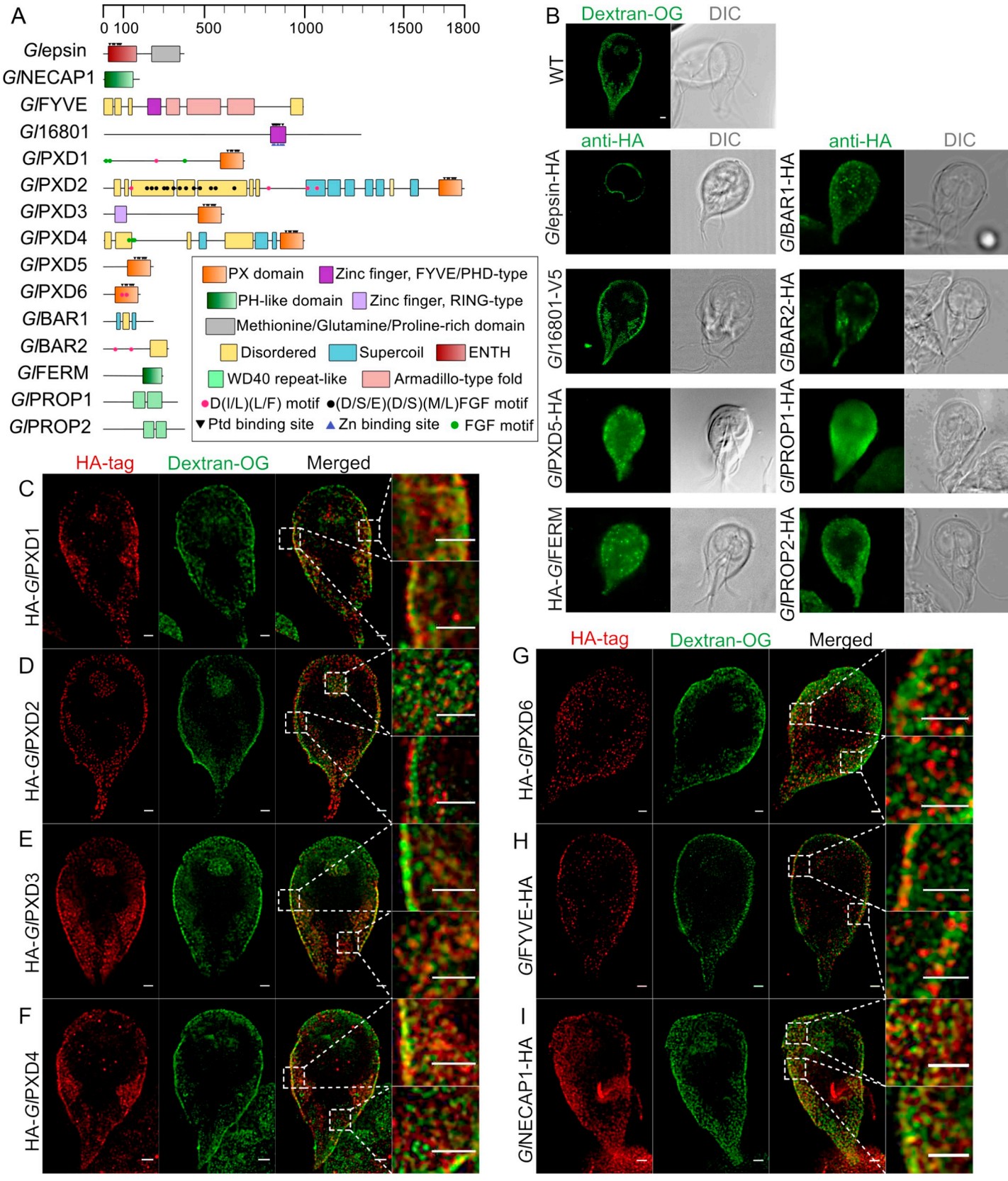

**Fig 1. Functional domain prediction analysis and subcellular localization of *G. lamblia* PIP-binding proteins.** (A) Predicted functional domains for all identified PIP-binding proteins including positions of repetitive motifs and putative lipid and Zn -binding residues using HHPRED, HMMER and InterProScan. Ptd– Phosphatidylinositol. (B) Conventional confocal light-microscopy analysis of representative non-transgenic trophozoites labelled with Dextran-OG (first panel) to mark PV lumina and of antibody-labelled trophozoites expressing HA-tagged PIP-binding protein reporters. Except for *Gl*epsin-HA and HA-*Gl*FERM, all tested reporter proteins localize in close proximity to peripheral vacuoles (PVs) at the cell cortex. Epitope-tagged HA-*Gl*PXD5 and *Gl*PROP1-HA additionally show signal distribution throughout the cell. Cells were imaged at maximum width, where nuclei and the bare-zone are at maximum diameter. Epitope-tagged *Gl*epsin-expressing cells were imaged at maximum width of the ventral disk. Scale bar: 1 μm. (C-I) Confocal STED microscopy analysis of trophozoites expressing epitope-tagged PIP-binding reporter proteins for *Gl*PXD1-6, *Gl*FYVE-HA and *Gl*NECAP1-HA (red channel) co-labelled with Dextran-OG as a marker for PV lumina (green channel). As shown in the merged insets, although all reporters are clearly PV-associated, reporters for proteins *Gl*FYVE-HA and HA-*Gl*PXD1 and 2 are proximal to the PM with respect to Dextran-OG, indicating they reside at the PV-PM interface. In contrast, reporters for HA-*Gl*PXD3 and *Gl*NECAP1-HA appear to intercalate PVs. Scale bars: 1 μm for full cell and inset images.

## PIP-binding proteins associated with clathrin assemblies present distinct lipid-binding profiles *in vitro*

PX domains [35] and FYVE [36–38] preferentially bind PI(3)P. Even though PH domains have rather promiscuous binding preferences, a subset of PH domains binds strongly to PtdIns(3,4,5)$P_3$ and PtdIns(4,5)$P_2$, as well as PtdIns(3,4)$P_2$ [39–41]. Based on the presence of conserved residues for lipid-binding in the giardial PXD1-6, FYVE and NECAP1 proteins (S1 Fig), we hypothesized that their lipid-binding preferences would also be conserved [20]. We tested this experimentally by expressing MBP-fused, epitope-tagged *Gl*PXD1-6, *Gl*FYVE and *Gl*NECAP1 lipid-binding domains (S2A and S2B Fig). Recombinant fusion proteins were affinity-purified and used in lipid binding assays either for commercially-available PIP gradients as membrane-supported arrays (1.56–100 pmol/spot) (Fig 2A) or membrane strips spotted with defined amounts (100 pmol/spot) of PIPs (S2C Fig). The negative control for binding consisted of a PIP array probed with purified epitope-tagged MBP alone, whereas the positive control consisted of a PIP array probed with a commercially-available anti-PI(4,5)$P_2$ antibody (Fig 2A).

Quantification of the chemiluminescence signals shows a marked preference of MBP-*Gl*PXD1 for PI(4,5)$P_2$ in PIP gradients (Fig 2B) which was corroborated by experiments using PIP strips (S2C Fig). Under these conditions, *Gl*PXD2, 3, and 6 show unexpectedly promiscuous binding preferences, with *Gl*PXD2 presenting a marked affinity for PI(3)P and PI(4,5)$P_2$, *Gl*PXD3 for PI(3)P and to a lesser extent PI(5)P, and *Gl*PXD6 for PI(3)P, PI(4)P, and PI(5)P (Fig 2B). These data were in line with results from independent PIP strip experiments (S2C and S2D Fig). MBP-*Gl*PXD4 and MBP-*Gl*PXD5 binding preferences could only be probed using PIP strips (S2C Fig), showing in both cases a marked affinity for PI(3,5)$P_2$ and PI(4,5)$P_2$ (S2C and S2D Fig). Binding preferences for MBP-*Gl*FYVE could not be determined, given that no signal was ever obtained on both PIP arrays and strips (Fig 2A, S2C and S2E Fig). Surprisingly, testing of *Gl*NECAP1 consistently detected cardiolipin as the preferred lipid moiety (Fig 2C; S2E Fig), with no detectable preference for PIP residues (S2C Fig). Taken together, our data show clearly distinguishable lipid binding profiles *in vitro*, with varying degrees of promiscuity for different PIP-binding domains.

## Depletion of free PI(3)P, PI(4,5)$P_2$ and PI(3,4,5)$P_3$, but not PI(4)P binding sites *in vivo* inhibits PV-mediated uptake of a fluid-phase marker

The marked preference of *Gl*PXD1-6 for PIP residues PI(3)P and PI(4,5)$P_2$ raised the question whether their perturbation would elicit loss-of-function phenotypes in fluid phase uptake by *Giardia* trophozoites. Using a combination of commercially available antibodies, heterologous reporter constructs and chemical treatment, we reduced the bioavailability of PI3P, PI(4,5)$P_2$, and in addition PI(3,4,5)$P_3$ and PI(4)P.

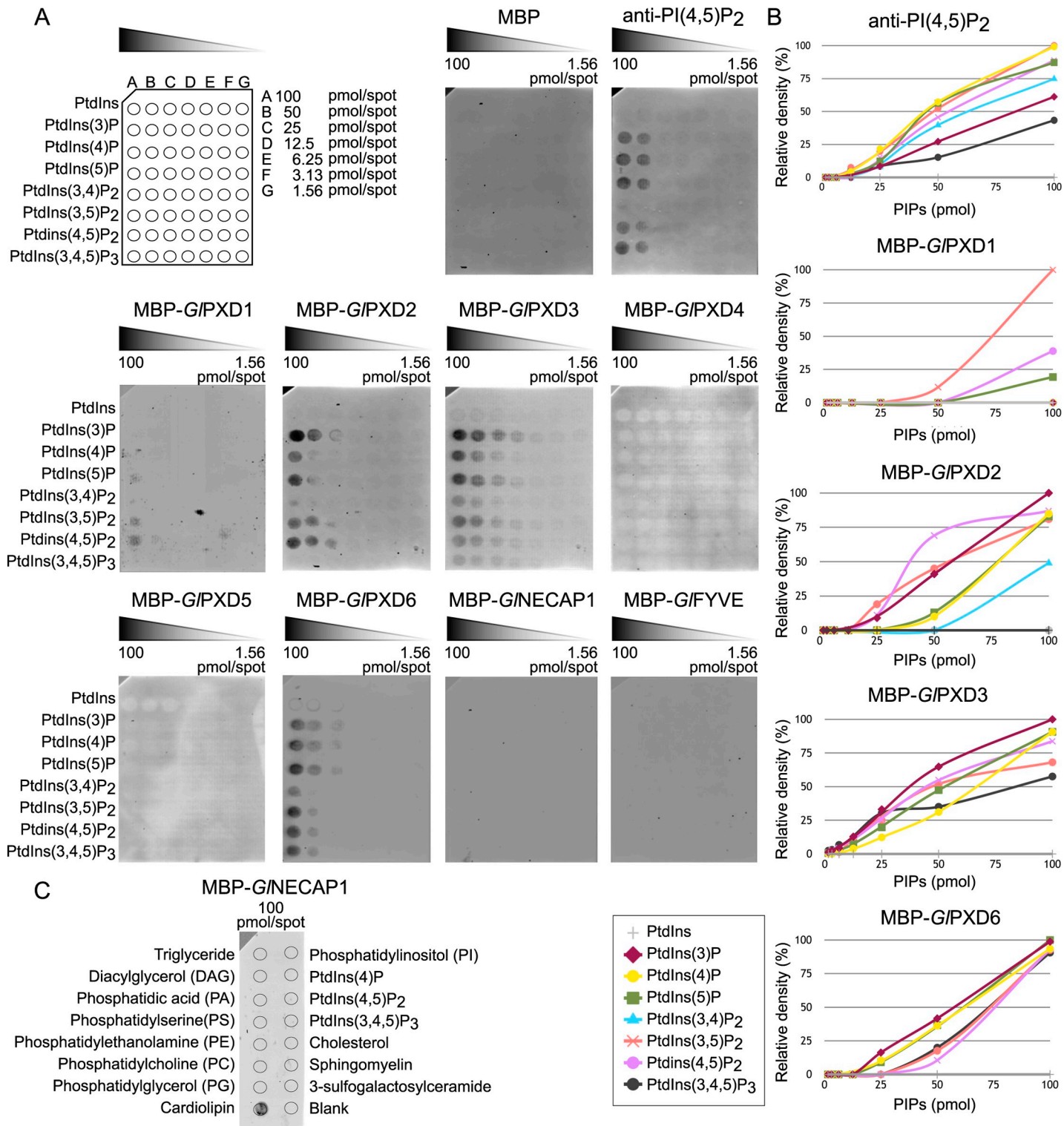

**Fig 2. Lipid-binding properties of selected giardial PIP-binding domains.** (A) Membrane-supported lipid arrays spotted with gradients of different phosphorylated variants of phosphatidylinositol (PtdIns), from 100pmol (A) to 1.56pmol/spot (G), were probed with fixed amounts (2.5 μg) of clathrin assemblies-associated epitope-tagged PIP-binding domains from *Gl*PXD1-6, *Gl*NECAP1 and *Gl*FYVE, followed by immunodetection of the epitope tag. The protein fusion partner MBP (MBP alone) and for antibodies raised against PI(4,5)P₂ (anti-PI(4,5)P₂) were included as negative and positive controls for binding, respectively. No signal using arrays was obtained for MBP-*Gl*PXD4 and MBP-*Gl*PXD5 however, binding preferences for these fusions were determined using lipid strips (S2A and S2B Fig). (B) Plots of densitometric

analyses using FIJI for each MBP-fused PIP-binding domain and each spotted PI/PIP residue based on array data presented in (A). (C) Testing of the binding affinity of the MBP-fused PIP-binding domain from *Gl*NECAP1 on a wider range of lipid residues detects cardiolipin as the preferred substrate.

Detection of PI(3)P, PI(4,5)P$_2$, and PI(3,4,5)P$_3$ in chemically fixed trophozoites by immunofluorescence microscopy with primary PIP-targeted antibodies highlights enrichment for all PIP moieties in the cortical region containing PVs (S3 Fig).

Ectopic expression of fluorescent high-affinity reporters for PI(3)P and PI(4)P, namely 2xFYVE-GFP and GFP-P4C [42], respectively, in transgenic *G. lamblia* trophozoites was used to identify membranes enriched for PI(3)P and PI(4)P deposition (Fig 3A–3D). Live microscopy of cells expressing 2xFYVE-GFP shows distinct reporter accumulation in cortical areas consistent with binding to PV membranes (Fig 3B, green panels), whereas representative cells from line GFP-P4C show a more diffused cytosolic staining pattern, with some accumulation at PVs (Fig 3D, green panels). Fluid-phase uptake of Dextran-R was assessed in cells from both transgenic lines, and compared to wild-type cells using quantification of signal intensity. Wild-type control cells and transgenic cells weakly expressing 2xFYVE-GFP (Fig 3A) incorporated large amounts of Dextran-R (Fig 3E). Conversely, a strong 2xFYVE-GFP signal correlated with low amounts of endocytosed Dextran-R detected at the cell periphery and with noticeably enlarged cells (Fig 3B). In contrast, there was no detectable difference in either Dextran-R uptake efficiency (based on fluorescent signal intensity) or cell width between weak (Fig 3C) and strong expressors (Fig 3D) of the GFP-P4C line. Cell width (Fig 3F) and fluid-phase uptake (Fig 3G) aberrant phenotypes in 2xFYVE-GFP cells were recorded with respect to wild-type control and GFP-P4C cells and tested for significance (p>0.05) on 100 cells/line selected in an unbiased fashion. These data translate into a significant negative correlation between expression of the PI(3)P-binding 2xFYVE-GFP reporter and fluid-phase uptake (Fig 3H) whereas only a slight albeit insignificant correlation was found between Dextran uptake and GFP-P4C expression (Fig 3I). Furthermore, performance of the Mann-Whitney test on our data confirmed the significance of the observed changes in cell width only in 2xFYVE-GFP-expressing cells, compared with non-transgenic cells. Specifically, the null hypothesis for no change in width was rejected for the WB vs 2xFYVE-GFP comparison (p(0.05)-value = 2.22045e-16) and accepted for the WB vs GFP-P4C comparison (p(0.05)-value = 0.182377).

The cationic antibiotic neomycin binds tightly to the headgroup of phosphoinositides with a marked preference for PI(4,5)P$_2$ and, to a lesser extent, PI(3,4,5)P$_3$ [43, 44]. As a means to perturb PI(4,5)P$_2$ and PI(3,4,5)P$_3$ availability in *Giardia* trophozoites, we tested its effect on fluid-phase uptake by treating wild-type trophozoites with 7.2 mM neomycin followed by uptake of Dextran-R. Quantitative light microscopy image analysis revealed a significantly lower level of Dextran-R in treated trophozoites (p<0.05) (Fig 3J and 3K) which remained vital and motile in the presence of neomycin up to 15mM for 50 minutes (S1–S4 Videos). Taken together, the data indicate that depletion of free binding sites for PI(3)P, PI(4,5)P$_2$, and PI(3,4,5)P$_3$, but not PI(4)P significantly impacts fluid-phase endocytosis through *G. lamblia* PVs.

## Functional characterization of *Gl*PXD1-4 and 6, *Gl*FYVE and *Gl*NECAP1

Manipulation of PIP residue homeostasis elicited PV-dependent fluid-phase uptake phenotypes. We hypothesized that changing expression levels of giardial PIP-binding proteins previously identified in clathrin interactomes would elicit aberrant uptake phenotypes in *Giardia* trophozoites. In addition, we explored the functional boundaries of each PIP-binding module by defining their protein interactomes. To test this, we used the previously-generated epitope-

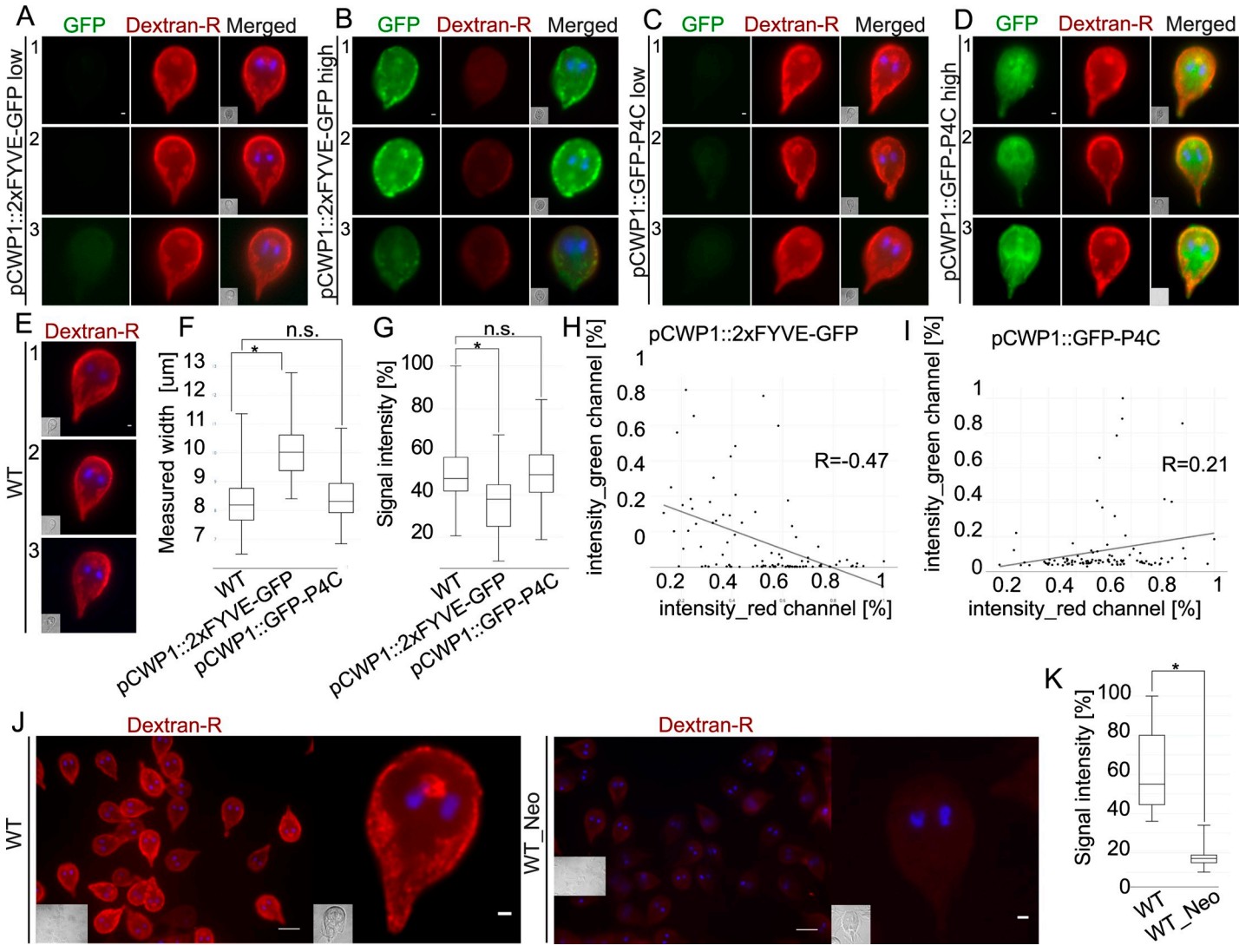

**Fig 3. Depletion of free PI(3)P, PI(4,5)P₂ and PI(3,4,5)P₃ binding sites in *G. lamblia* trophozoites elicits uptake and morphological phenotypes.** (A-D) Light microscopy-based immunofluorescence analysis of representative transgenic trophozoites expressing *Legionella*-derived PIP-binding constructs. (A-B) Compared to low 2xFYVE-GFP-expressing cells from the same population, reduction of PI(3)P binding sites in cells highly expressing a regulated encystation-dependent epitope-tagged construct 2xFYVE-GFP (GFP) inhibits uptake of fluid-phase marker Dextran-R. Scale bars: 1 μm. (C-D) Expression levels of PI(4)P-binding epitope-tagged construct GFP-P4C expression (GFP) have no visible impact on Dextran-R signal at PVs of transfected cells. Scale bars: 1 μm. (E) Dextran-R uptake in non-transgenic wild-type cells as negative controls for construct-induced uptake phenotypes. Scale bars: 1 μm (F) Box-plot representing the distribution of cell width (in μm) across at least 100 wild-type, 2xFYVE-GFP- and GFP-P4C- expressing cells selected in an unbiased fashion. A statistically significant (two-sided t-test assuming unequal variances, p<0.05) increase in median cell width with respect to non-transgenic cells is detected for 2xFYVE-GFP- but not GFP-P4C- expressing cells. Asterisks indicate statistical significance. n.s.: not significant. (G) Box-plot representing the distribution of measured Dextran-R signal intensity across at least 100 wild-type, 2xFYVE-GFP- and GFP-P4C- expressing cells selected in an unbiased fashion. A statistically significant (two-sided t-test assuming unequal variances, p<0.05) decrease in Dextran-R signal intensity, normalized to wild-type cells (100%), is detected for 2xFYVE-GFP- but not for GFP-P4C- expressing cells. Asterisks indicate statistical significance. n.s.: not significant. (H) A statistically significant (p<0.5) linear correlation exists between Dextran-R signal (*x*-axis, intensity_red channel [%]) and 2xFYVE-GFP expression (*y*-axis, intensity_green channel [%]) measured across 100 cells. (I) The apparent linear correlation between GFP-P4C expression (*y*-axis, intensity_green channel [%]) and Dextran-R signal (*x*-axis, intensity_red channel [%]) is not statistically significant (p<0.5). (J) Wide-field microscopy-based immunofluorescence analysis of the impact of neomycin treatment on Dextran-R uptake to deplete PI(4,5)P₂ binding sites in non-transgenic wild-type cells. With respect to non-treated cells (WT; left panel), Dextran-R signal at PVs is visibly impacted in non-transgenic neomycin-treated cells (WT_Neo; right panel). Scale bars: 10 μm for full wide-field image, 1 μm for a single cell. (K) Box-plot representing the distribution of measured Dextran-R signal intensity across 100 wild-type cells, either untreated (WT) or treated with neomycin (WT_Neo). Neomycin treatment causes a statistically significant (two-sided t-test assuming unequal variances, p<0.05) decrease in Dextran-R signal. Scale bars: wide-field: 10 μm; single cells: 1 μm. For all images, nuclei are labelled with DAPI (blue). Insets: DIC images.

**Table 2. Overview of interactomes derived from epitope-tagged reporter lines for full-length *Gl*PXD1-4 and 6, *Gl*FYVE and *Gl*NECAP1.** Main putative interaction partners are highlighted for each antibody-based affinity co-IP experiment. The type of interaction detected (reciprocal/one-sided) and its relative strength as indicated by the number of exclusive spectral counts associated to each candidate interactor, are also included.

| Bait-ORF number | Partners | Interaction | Data | Annotation and reference(Zumthor et al., 2016 if not indicated) |
|---|---|---|---|---|
| *Gl*PXD1 | GL50803_102108 | Reciprocal—strong | S4A Fig | *Gl*CHC |
| GL50803_7723 | | | | |
| | GL50803_4259 | Reciprocal—strong | S4A Fig | *Gl*CLC |
| | GL50803_21423 | Reciprocal—strong | S4A Fig | *Gl*AP2-β |
| | GL50803_17304 | Reciprocal—weak | S4A Fig | *Gl*AP2-α |
| | GL50803_8917 | One sided—strong | S4A Fig | *Gl*AP2-μ |
| | GL50803_14373 | One sided—weak | S4A Fig | *Gl*DRP |
| | GL50803_16595 | One sided—weak | S4A Fig | *Gl*PXD2 |
| *Gl*PXD2 | GL50803_102108 | Reciprocal—strong | S4A Fig | *Gl*CHC |
| GL50803_16595 | GL50803_4259 | One sided—weak | S4A Fig | *Gl*CLC |
| | GL50803_21423 | Reciprocal—strong | S4B Fig | *Gl*AP2-β |
| | GL50803_17304 | Reciprocal—strong | S4B Fig | *Gl*AP2-α |
| | GL50803_8917 | One sided—strong | S4B Fig | *Gl*AP2-μ |
| | GL50803_5328 | One sided—strong | S4B Fig | *Gl*AP2-σ |
| | GL50803_42357 | Reciprocal weak | S4B Fig | *Gl*PXD4 |
| | GL50803_17195 | One sided—weak | S4B Fig | *Gl*NECAP1 (this study) |
| *Gl*PXD3 | GL50803_102108 | Reciprocal—strong | S4C Fig | *Gl*CHC |
| GL50803_16596 | GL50803_21423 | Reciprocal—strong | S4C Fig | *Gl*AP2-β |
| | GL50803_17304 | Reciprocal—strong | S4C Fig | *Gl*AP2-α |
| | GL50803_14373 | One sided—weak | S4C Fig | *Gl*DRP |
| | GL50803_15411 | Reciprocal—strong | S4C Fig | *Gl*15411 |
| | GL50803_9606 | One sided—weak | S4C Fig | *Gl*9605 (this study) |
| *Gl*PXD4 | GL50803_102108 | One sided—strong | S4A Fig | *Gl*CHC |
| GL50803_42357 | GL50803_21423 | One sided—weak | S4A Fig | *Gl*AP2-β |
| | GL50803_14373 | One sided—strong | S4A Fig | *Gl*DRP |
| | GL50803_16595 | Reciprocal weak | S4A Fig | *Gl*PXD2 (this study) |
| *Gl*PXD6 | GL50803_102108 | One sided—strong | S4A Fig | *Gl*CHC |
| GL50803_24488 | GL50803_21423 | One sided—strong | S4A Fig | *Gl*AP2-β |
| | GL50803_14373 | One sided—strong | S4A Fig | *Gl*DRP |
| | GL50803_16596 | One sided—weak | S4A Fig | *Gl*PXD3 (this study) |
| | GL50803_16653 | One sided—weak | S4A Fig | *Gl*FYVE (this study) |
| *Gl*FYVE GL50803_16653 | GL50803_102108 | Reciprocal—strong | S4D Fig | *Gl*CHC |
| | GL50803_14373 | Reciprocal—strong | S4D Fig | *Gl*DRP |
| *Gl*NECAP1 | GL50803_102108 | One sided—strong | S4E Fig | *Gl*CHC |
| GL50803_17195 | GL50803_17304 | One sided—strong | S4E Fig | *Gl*AP2-α |
| | GL50803_21423 | One sided—strong | S4E Fig | *Gl*AP2-β |
| | GL50803_14373 | One sided—strong | S4E Fig | *Gl*DRP |

tagged reporter lines for full-length *Gl*PXD1-4 and 6, *Gl*FYVE and *Gl*NECAP1 (Fig 1C–1I) for assessing the effects of ectopic expression on fluid-phase uptake phenotypes. Furthermore, we used the same lines as a source of tagged "baits" in antibody-based affinity co-immunoprecipitation (co-IP) and identification of reporter-associated protein complexes (Table 2). Further investigation of *Gl*PXD5 was abandoned at this stage due to its intractably low levels of expression.

## The extended interactomes of *Gl*PXD1, *Gl*PXD4 and *Gl*PXD6

Epitope-tagged, full-length *Gl*PXD1 is a validated *Gl*CHC interaction partner; its extended interactome confirms association to all core clathrin assembly components (*Gl*CHC, *Gl*CLC, *Gl*DRP, and *Gl*AP2) (Figs 4 and S4A and S1 Table) [19]. A weaker interaction with *Gl*PXD2 was also found. The *Gl*PXD4 interactome includes *Gl*CHC and *Gl*DRP and, uniquely for the *Gl*PXD protein family, a previously confirmed interaction with *Gl*PXD2 [19] albeit detected at lower stringencies (95_2_95, 2 hits) (Figs 4 and S4A and S4 Table). A putative SNARE protein GL50803_5785, previously identified in the *Gl*Tom40 interactome [45], was detected at lower stringencies (95_2_95, 2 hits). Similar to *Gl*PXD1, *Gl*PXD6 showed strong interaction with the β subunit of *Gl*AP2 and *Gl*CHC (Figs 4 and S4A), although the reverse interaction was not detected in the previously-published clathrin-centered interactome [19]. Using lower stringency parameters (95_2_50, 3 hits), revealed interaction with *Gl*FYVE, *Gl*PXD3 and *Gl*DRP (Figs 4 and S4A and S5 Table). The *Gl*PXD6 interactome includes *Gl*16717, a protein of unknown function predicted to carry a StAR-related lipid-transfer domain (Steroidogenic Acute Regulatory protein, START) domain [46]. Ectopic expression of epitope-tagged *Gl*PXD1, 4 and 6 elicited no discernible PV-related phenotypes.

**Ectopic expression of tagged *Gl*PXD2 severely perturbs PV organization.**   Mining the *Gl*PXD2 protein interactome dataset with high stringency parameters confirmed interactions with *Gl*CHC, *Gl*AP2, and *Gl*PXD4 (Figs 4 and S4B and S2 Table). Furthermore, we identified three predicted SNARE proteins: *Gl*5785, GL50803_14469 (*Gl*14469; at lower stringencies 95_2_50, 9 hits), and GL50803_10013 (*Gl*10013; Figs 4 and S4B) [47]. The SNARE *Gl*5785 was detected also in the interactomes of *Gl*PXD4 and *Gl*TOM40 [45]. *Gl*NECAP1 was also identified as a *Gl*PXD2 interacting partner, albeit only by applying low stringency parameters (95_2_50, represented by a dashed line, Figs 4 and S4B).

In contrast to ectopic expression of tagged *Gl*PXD1, 4, and 6, expression of an epitope-tagged reporter HA-*Gl*PXD2 elicited a distinct phenotype. In contrast to non-transgenic wild-type cells (Fig 5A) and weakly-expressing HA-*Gl*PXD2 cells (Fig 5B–5D upper panels), gated STED imaging of trophozoites strongly expressing HA-*Gl*PXD2 showed large membranous clusters which also accumulated Dextran-R (Fig 5B) and were bound by both anti-*Gl*CHC (Fig 5C) and anti-PI(3)P (Fig 5D) antibodies.

Transmission electron microscopy (tEM) analysis confirmed the presence of randomly distributed peripheral PV clusters in cells expressing HA-*Gl*PXD2 (Fig 5E; left panel) which were not present in representative wild-type control cells (Fig 5E; right panel).

**The *Gl*PXD3 interactome is connected to clathrin assemblies and includes a novel dynamin-like protein.**   *Gl*DRP, *Gl*CHC, and *Gl*AP2 (α/β subunits) were detected in the *Gl*PXD3 interactome, thereby establishing the association of this PX domain protein with clathrin assembly structures at the PV/PM interface (Figs 4 and S4C and S3 Table). A pseudokinase (*Gl*15411 [48]) previously identified in *Gl*CHC assemblies was also found in the *Gl*PXD3 interactome (Figs 4 and S4C [19]). Furthermore, the *Gl*PXD3 and *Gl*15411 interactomes share proteins GL50803_16811 (*Gl*16811) tentatively annotated as a ZipA protein in GDB, and proteins GL50803_87677 (*Gl*87677) and GL50803_17060 (*Gl*17060), annotated as a NEK kinase and an ankyrin-domain carrying protein, respectively (Figs 4 and S4C). Unique interaction partners for *Gl*PXD3 include the SNARE protein *Gl*7309 [47] and *Gl*NSF (GL50803_114776) [49]. In addition, protein GL50803_103709 carrying a predicted N-terminal BRO domain and protein GL50803_9605 were identified as unique *Gl*PXD3 interaction partners (Figs 4 and S4C). Furthermore, the StAR-related lipid-transfer protein *Gl*16717, already found in the *Gl*PXD6 interactome was also found to be a low-stringency interaction partner for *Gl*PXD3 and *Gl*15411, thereby connecting the *Gl*PXD3 and *Gl*PXD6-*Gl*FYVE circuits.

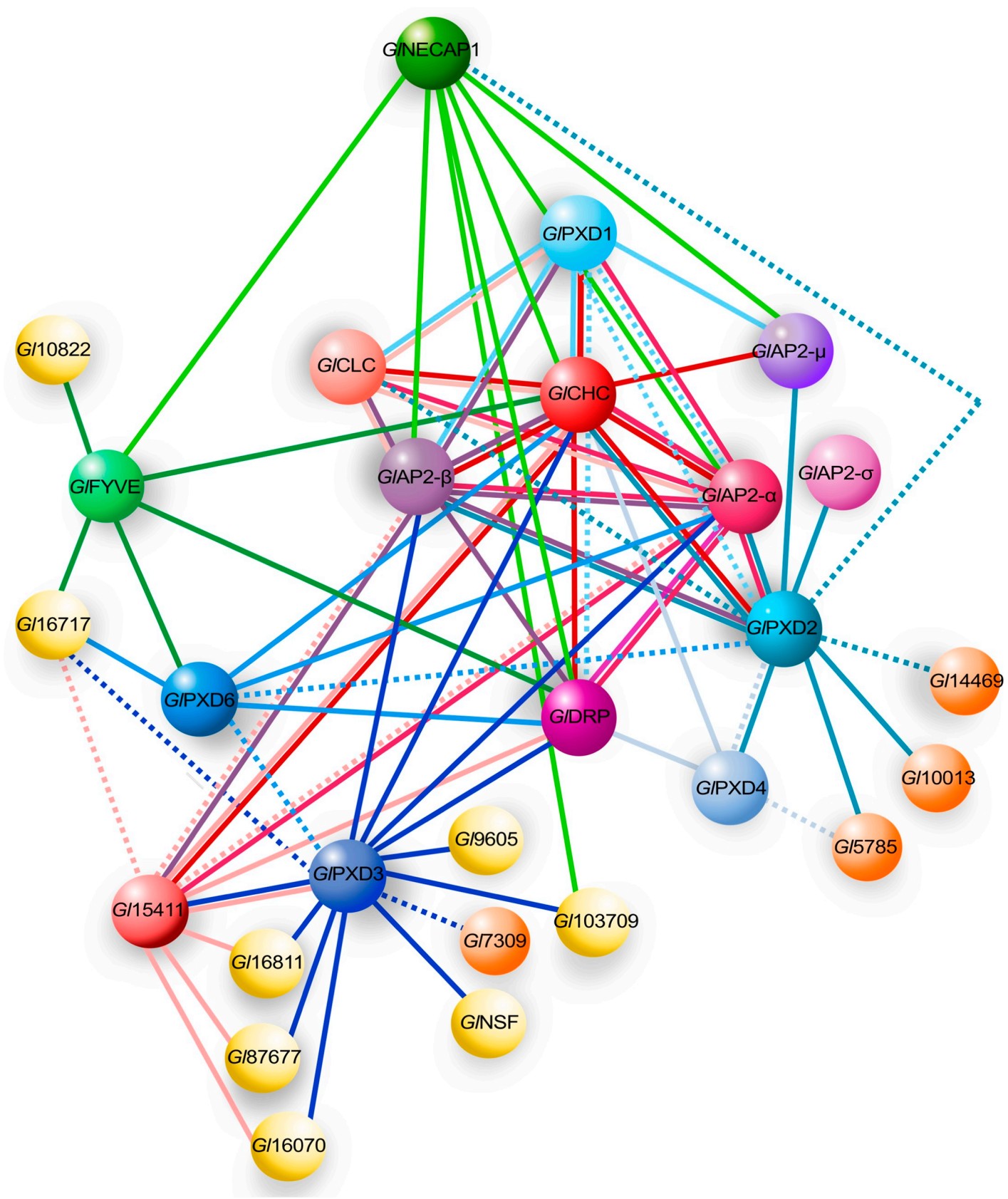

**Fig 4. Overview of core protein interactomes determined from co-IP analyses.** Interactomes for *Gl*FYVE, *Gl*NECAP1 and *Gl*PXD3, 4, and 6 defined by co-IP analysis were integrated with previously published data [19] for core clathrin assembly, *Gl*PXD1, *Gl*PXD2 and *Gl*FYVE interactomes. Solid lines: interactions detected at high stringency. Dashed lines: interactions detected at low stringency. Yellow partners are currently annotated on GDB as "hypothetical protein" *i.e.* proteins of unknown function.

Protein *Gl*9605, the sixth most abundant hit in the *Gl*PXD3 interactome (S3 Table), and currently annotated as having an unknown function, was localized in close proximity to PVs (Fig 6A) and identified as a highly-diverged dynamin-like protein (Fig 6B).

In support of this, the predicted GTPase domain in *Gl*9605 contains signature motifs in the P-loop (G1), switch 1 (G2) and switch 2 (G3) regions [50–52]. Conserved motifs in the G4 region are only partially maintained (Fig 6B). To test residue conservation on a structural level, *Gl*9605 was subjected to *ab initio* modelling using I-TASSER and the resulting tertiary structure was superimposed on that of a dynamin-like protein 2 (DLP2 Cj:5ovW) [53], *Gl*9605's closest structural homologue (Fig 6C). A structural overlap TM-score of 0.913 suggests an almost perfect structural match, with clear chemical and positional conservation of key residues involved in GTPase activity (Fig 6C). We sought to elicit a dominant-negative phenotype by engineering *Gl*9605 K73E and S74N mutants [54]. In contrast to either wild-type cells or cells expressing a wild-type epitope-tagged *Gl*9605 control, expression of *Gl*9605 K73E and S74N mutant reporters inhibited fluid-phase uptake of Dextran-R in a statistically significant manner (p<0.05; Fig 6D).

**Regulated ectopic expression of *Gl*FYVE variants inhibits fluid-phase uptake.** *Gl*FYVE is a confirmed interactor of clathrin assemblies [55] through specific association to *Gl*CHC and *Gl*DRP (Figs 4 and S4D and S6 Table). *Gl*FYVE's extended interactome includes *Gl*PXD6 and *Gl*NECAP1.

To characterize the function of *Gl*FYVE and to test whether a dominant-negative effect on uptake could be elicited, we performed a deletion analysis by generating epitope-tagged C-terminal (pCWP1::NT-*Gl*FYVE-HA) and N-terminal (pCWP1::CT-*Gl*FYVE-HA) truncation constructs. These consist of either the disordered region followed by the FYVE domain (Fig 7A), residues 1–300 or the armadillo repeat-rich (ARM repeats) domains (Fig 7A), residues 301–990), respectively.

Expression of both constructs is regulated by an inducible promoter which is de-repressed during transient induction of encystation, the process during which a flagellated trophozoite differentiates to a cyst [56]. After a short (6h) induction pulse, transfected cells were subjected to Dextran-R uptake. In cells expressing full-length pCWP1::*Gl*FYVE-HA and truncated variants, the amount of Dextran-R accumulated in PVs was significantly (p<0,05) lower (Fig 7B, box plot). Furthermore, IFA analysis of pCWP1::NT-*Gl*FYVE-HA cells revealed the presence of structures which overlapped neither with Dextran-R-labelled PVs (Fig 7B) nor with encystation specific vesicles (ESVs) labeled with the anti-CWP1 antibody (Fig 7C). In contrast, CT-*Gl*FYVE-HA and full length *Gl*FYVE-HA localized predominantly to PVs (Fig 7B–7D). The subcellular localization of *Gl*CHC in these lines and in a wild-type control overlapped with the truncated CT-*Gl*FYVE-HA variant, but only partially with NT-*Gl*FYVE-HA and *Gl*FYVE-HA (Fig 7D).

**Ectopic expression of *Gl*NECAP1 significantly impairs fluorescent Dextran uptake.** Co-IP using epitope-tagged *Gl*NECAP1 confirmed interaction with clathrin assembly components *Gl*AP2-β, μ and α subunits, *Gl*CHC and *Gl*DRP. Interaction with *Gl*FYVE (S4D Fig) and, at lower stringency also for *Gl*PXD2 (S4B Fig), could be confirmed (Figs 4 and S4E and S7 Table).

Three putative conserved AP2-interacting motifs were identified using multi-sequence alignment; the high affinity WxxF motif at the N-terminus, two residues being invariant

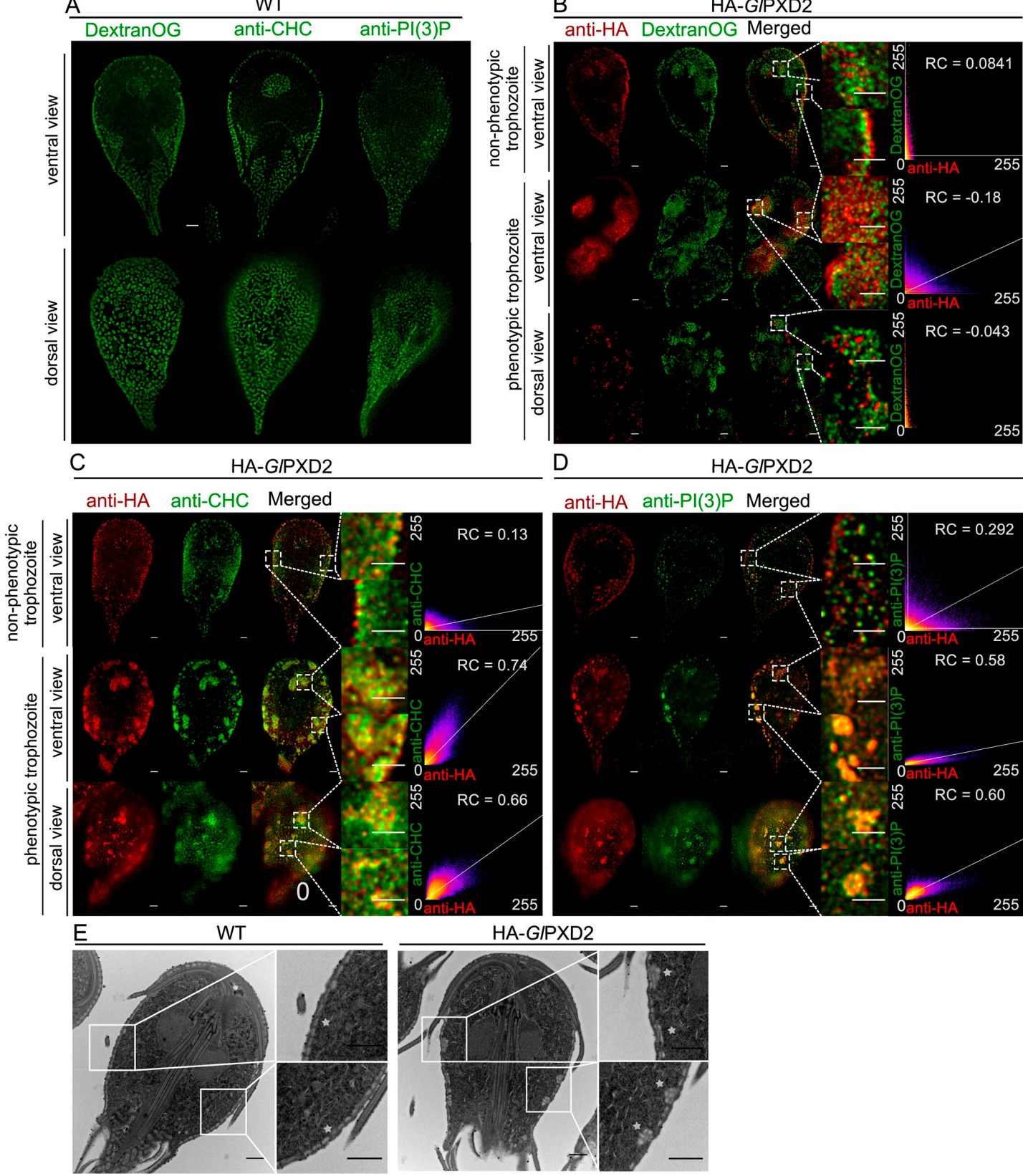

**Fig 5. The impact of *Gl*PXD2 ectopic expression on PV morphology.** (A) STED-microscopy-based immunofluorescence analysis of ventral (upper panel) and dorsal (lower panel) views of representative non-transgenic wild-type cells labelled with Dextran-OG, anti-*Gl*CHC (*Gl*CHC) and anti-PI(3)P antibodies (PI(3)P) detects association of all markers to PVs. Insets: DIC images. Scale bars: 1 μm (B) STED-microscopy based immunofluorescence analysis of transgenic cells either weakly (upper panel, non-phenotypic trophozoite) or strongly (lower panels, phenotypic trophozoite) expressing an epitope-tagged *Gl*PXD2 variant (HA-*Gl*PXD2) and subjected to Dextran-OG uptake detects co-labelled PV-derived peripheral clusters which are also detected with (C) anti-*Gl*CHC (*Gl*CHC) and (D) anti-PI(3)P (PI(3)P) antibodies. Scale bars: single cells: 1 μm; merged insets: 1 μm. (E) tEM experiments detect peripheral PV-associated membranous clusters in cells expressing a tagged *Gl*PXD2 variant (HA-*Gl*PXD2; upper panel and insets) which were not detected in non-transgenic cells (WT; lower panel and insets). Scale bars: single cells: 1 μm; insets: 1 μm.

throughout evolution, K147 and G149, and AP2-beta linker interacting residues binding sites (Fig 8A) [57].

*De novo* 3D modelling confirms overall structural conservation of all key residues in *Gl*NE-CAP1 compared to mammalian NECAP1 (Fig 8B). Furthermore, the interacting interface of NECAP1 with the β-linker region of AP2 was also identified in the structural model for *Gl*NE-CAP1 (Fig 8B).

To test whether expression of a *Gl*NECAP1 variant lacking the putative high-affinity motif WVIF could elicit a dominant-negative uptake effect, a deletion construct *Gl*NECAP1ΔW-VIF-HA was synthesised (Fig 8A) for ectopic expression. Accumulation of Dextran-R into PVs detected by microscopy was significantly lower (p<0.05) in transgenic cells ectopically expressing *Gl*NECAP1-HA or an APEX- and epitope- tagged variant *Gl*NECAP1-APEX2-2HA compared with wild type controls (Fig 8C, box plot). Conversely, ectopic expression of a deletion construct *Gl*NECAP1ΔWVIF-HA (Fig 8C, *Gl*NECAP1ΔWVIF-HA) had no discernible effect on accumulation of Dextran-R in PVs (Fig 8C, box plot). Ectopic expression of the genetically encoded enzymatic reporter [58, 59] *Gl*NECAP1-APEX2-2HA was associated to significantly enlarged PVs in tEM compared to wild type controls (Fig 8D; S5 Fig).

### *Gl*PXD3 associates specifically to PVs as membrane coat

Co-localization studies with Dextran-OG and ectopically expressed HA-*Gl*PXD3 show apparent coating of the entire PV membrane on the cytoplasmic side by the reporter construct (Fig 9A).

This provided us with an opportunity to generate measurements of PV organelles in optical sections using 3D STED microscopy followed by reconstruction and rendering with IMARIS. Rendered images show hive-like *Gl*PXD3-labelled structures predominantly in the cortical area of the cell underneath the PM that clearly surround the entire PV membrane (Fig 9B). The major and minor principal axes of these structures measured 437 +/- 93 nm and 271 +/- 60 nm. Consistent with the subcellular localization of this marker on the cytoplasmic side of PV membranes, these values were significantly higher (p≤ 0.05) than those obtained from PVs labeled with Dextran-OG (371 +- 79 nm and 221 +/- 49 nm) (Fig 9C and 9D). Signal overlap of epitope-tagged *Gl*PXD3 with endogenous *Gl*CHC as a marker for the PM-PV interface [19] in fluorescence microscopy is low. The image data indicate that both labels have distinct distributions but may spatially overlap at focal clathrin assemblies in small areas at the PV-PM interface (Fig 9E). Similarly, labelling for both PI(3)P and a reporter *Gl*PXD3 variant showed minimal signal overlap (Fig 9F), despite the strong affinity of the latter for this lipid in *in vitro* lipid-array binding experiments (Fig 2A and 2B).

## Discussion

### PIPs and PIP binders in *G. lamblia*

PIPs are recognized spatiotemporal organizers and decorate the surface of the eukaryotic cell's plasma and endo–membrane system [1–3]. *G. lamblia* is no exception; despite its significant

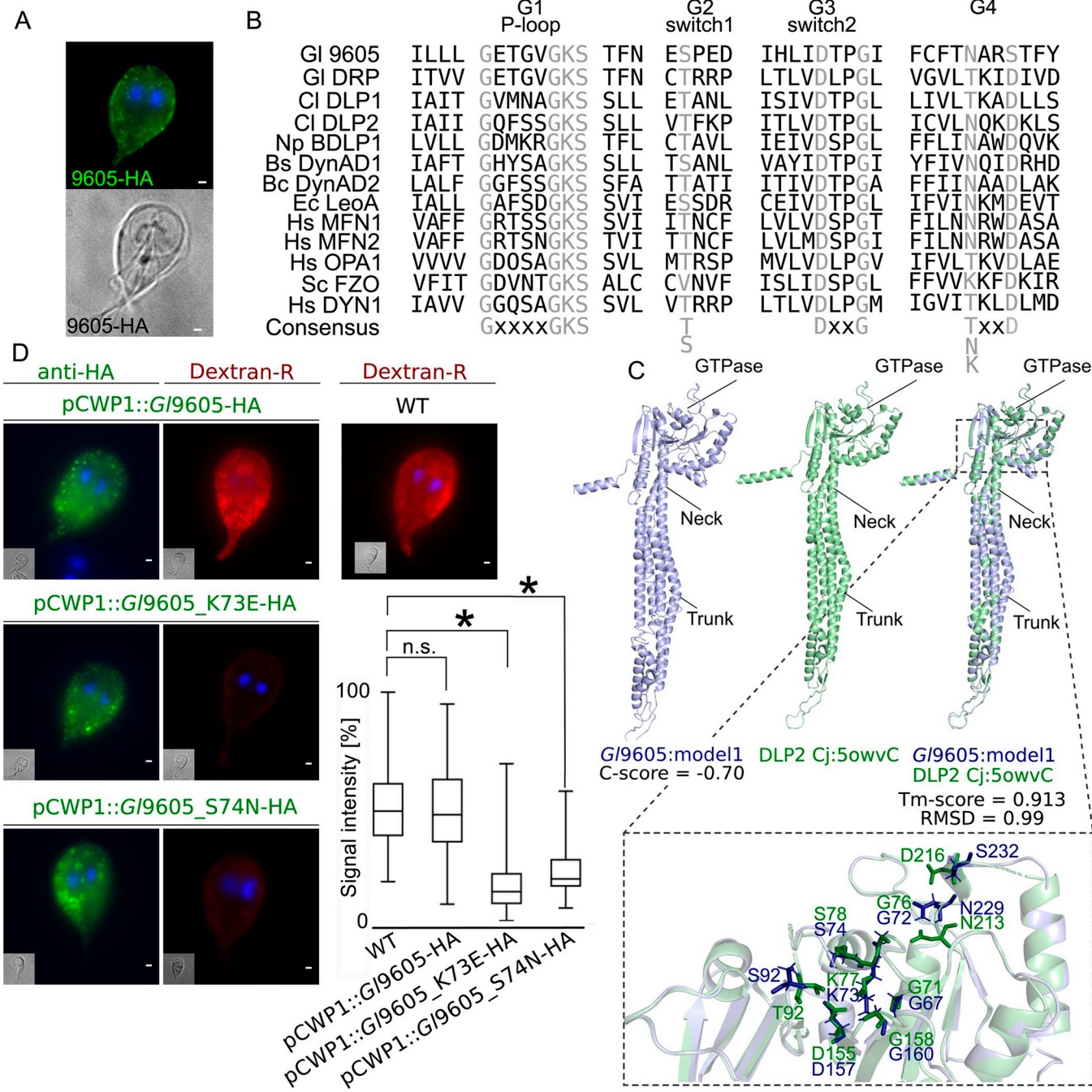

**Fig 6. The extended *Gl*PXD3 interactome includes a novel dynamin-like protein in *G. lamblia*.** (A) Light-microscopy-based immunofluorescence analysis of representative transgenic trophozoites expressing epitope-tagged reporter variants (green) for protein *Gl*9605. Cells were imaged at maximum width, where nuclei and the bare-zone are at maximum diameter. Nuclei are labelled with DAPI (blue). Scale bar: 1 μm (B) MSA analysis G1-P-loop, G2 switch 1, G3 switch 2 and G4 regions of the conserved GTPase domains of *Gl*9605, *Gl*DRP, *Campylobacter jejuni* DLP1 (Uniprot accession CJ0411) and DLP2 (CJ0412), *Nostoc punctiforme* BDLP1 (B2IZD3), *Bacillus subtilis* DynAD1 (P54159), *Bacillus cereus* DynAD2 (CUB17917), and *Escherichia coli* LeoA (E3PN25) bacterial dynamin-like proteins (BDLPs), *Homo sapiens* MFN1 (Q8IWA4), MFN2 (O95140), OPA1 (O60313) and DYN1 (Q05193), and *Saccharomyces cerevisiae* Fzo1p (P38297). Conserved positions are highlighted in grey. (C) I-TASSER *de novo* predicted 3D structure for *Gl*9605 (blue) and its closest known structural homologue, *C. jejuni* DLP2 (5owvC; green) indicating the GTPAse, neck and trunk regions that characterize BDLPs. A close-up view of the overlapping structures in the GTPase domains of *Gl*9605 (blue) and *C. jejuni* DLP2 (5owvC; green) marked with specific residues important for GTP binding and catalytic activity. (D) Quantitative microscopy-based immunofluorescence analysis of Dextran-R signal in cells expressing either a non-mutated full-length epitope-tagged *Gl*9605 or mutated *Gl*9605 K73E and S74N variants. In contrast to non-transgenic wild-type controls and *Gl*9605-HA expressing cells, expression of *Gl*9605 K73E and S74N variants inhibited Dextran-TxR uptake in a statistically significant fashion (box-plot). Asterisks indicate statistical significance. n.s.: not significant.

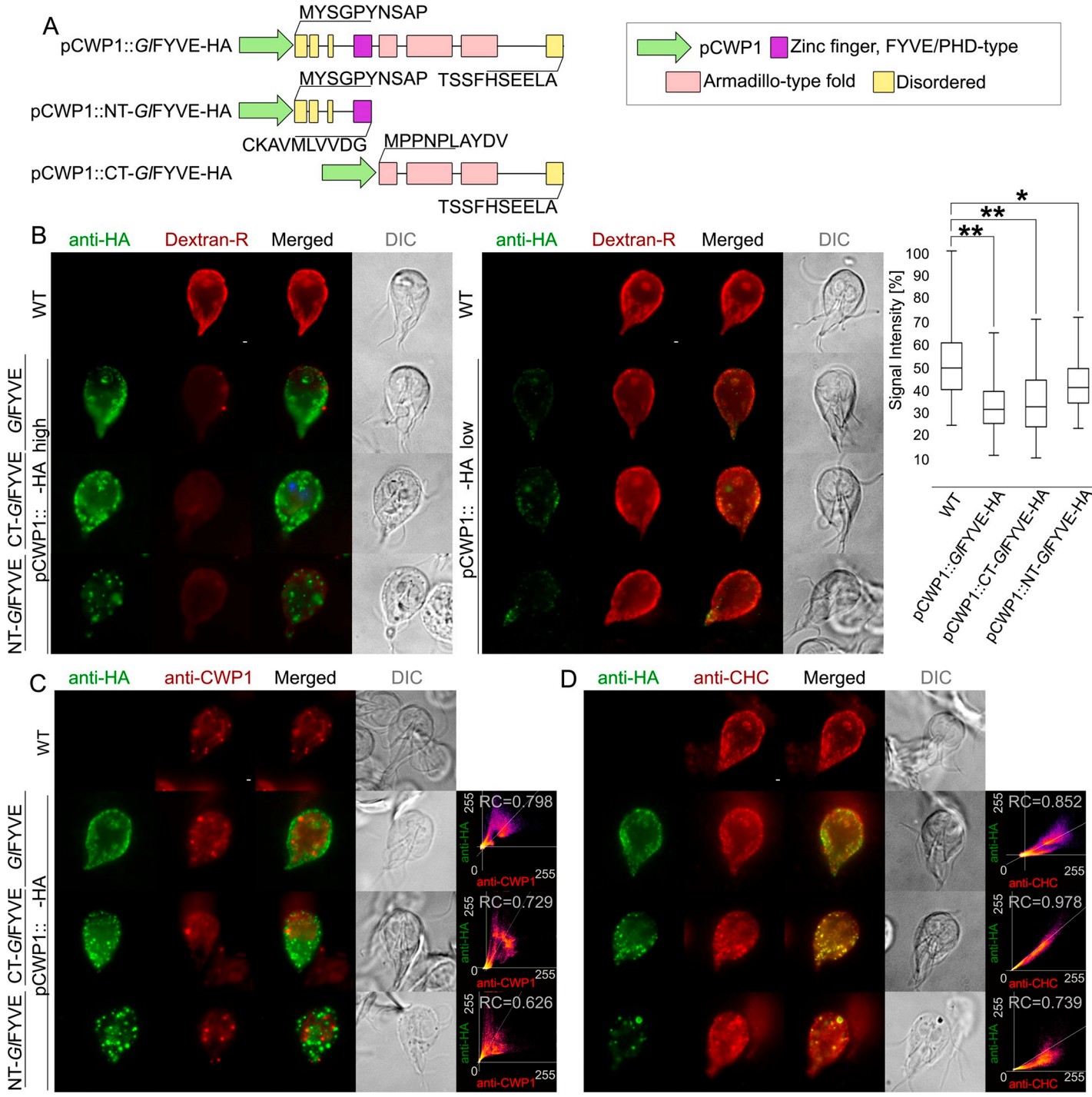

**Fig 7. Regulated ectopic expression of *Gl*FYVE variants inhibits fluid-phase uptake.** (A) C-terminally epitope-tagged full-length (top; pCWP1::*Gl*FYVE-HA), C-terminal truncated (middle; pCWP1::NT-*Gl*FYVE-HA, residues 1–300) and N-terminal truncated (bottom; pCWP1::CT-*Gl*FYVE-HA, 301–990 residues) constructs for *Gl*FYVE were generated for regulated expression and phenotype testing. (B) Confocal imaging and immunofluorescence analysis of 6hr encysting non-transgenic wild-type cells and cells expressing pCWP1-driven constructs *Gl*FYVE-HA, NT-*Gl*FYVE-HA or CT-*Gl*FYVE-HA (anti-HA) shows statistically significant differences in their ability to take up Dextran-R (two-sided t-test assuming unequal variances, p<0.05). Cells expressing construct pCWP1::NT-*Gl*FYVE-HA present additional structures that are not detected in other lines and do not associate with Dextran-R labelling. Asterisks indicate statistical significance: * p<0.05; ** p<0.005. n.s.: not significant. DIC: differential interference contrast. Scale bars: 1 μm. (C) Confocal imaging and immunofluorescence analysis of 6hr encysting non-transgenic wild-type cells and cells expressing pCWP1-driven constructs *Gl*FYVE-HA, NT-*Gl*FYVE-HA or CT-*Gl*FYVE-HA (anti-HA) using anti-CWP1-TxRed antibody (anti-CWP1) shows that the membrane compartments found in NT-*Gl*FYVE-HA-expressing cells are not related to encystation-specific vesicles. Scale bars: 1 μm. (D) Antibody-based immunofluorescence analysis of *Gl*CHC deposition (anti-CHC) in 6hr encysting non-transgenic wild-type cells and cells expressing pCWP1-driven constructs

*Gl*FYVE-HA, NT-*Gl*FYVE-HA or CT-*Gl*FYVE-HA (anti-HA) detects a significant degree of *Gl*CHC association to the CT-*Gl*FYVE-HA variant, with only partial association to NT-*Gl*FYVE-HA and *Gl*FYVE-HA constructs. Scale bars: 1 μm.

reduction in endomembrane complexity, this species maintains a variety of PIP residues, mostly located at the cell periphery. We identified 11 novel proteins, in most cases of unknown function that carry predicted PIP-binding modules and primarily localize in close proximity to PVs.

All hitherto identified PIP-binding proteins in *G. lamblia* can be loosely grouped in two categories; they are either relatively small proteins (up to 400 amino acid residues) consisting almost entirely of the PIP-binding module (e.g. *Gl*PXD6 and *Gl*NECAP1), or they are large proteins consisting of a single predicted domain for PIP-binding associated to domains of unknown function (e.g. *Gl*PXD2 and *Gl*FYVE). A full functional characterization of the latter is a challenge given the level of genomic sequence divergence in *G. lamblia*. This makes it currently difficult to determine whether sequences are lineage-specific or so diverged as to be unrecognizable orthologues of previously characterized proteins. Hence, structural annotation of large *G. lamblia* proteins carrying PIP-binding modules such as *Gl*PXD2 or *Gl*FYVE is limited to the lipid binding domain.

Eight out of 14 identified PIP-binding modules are either directly or indirectly associated to clathrin assemblies. Their PIP binding preferences, as measured using *in vitro* lipid-binding assays, are clearly distinct despite showing a varying degree of promiscuity, consistent with previously published data [20]. In contrast to previous reports, we could not measure PIP residue binding activity for *Gl*FYVE using *in vitro* lipid-binding assays [22]. Furthermore, *Gl*NE-CAP1 showed a distinctive and highly specific binding preference for cardiolipin. This is a surprising finding since cardiolipin is an abundant phospholipid of the inner mitochondrial membrane [60] whose presence in *Giardia* is controversial [61, 62]. Although *Gl*NECAP1 lacks canonical motifs for cardiolipin binding [63], previous reports on the identification of cardiolipin-binding PH domains [64, 65] lend support to the observation that the PH-like domain in *Gl*NECAP1 could bind cardiolipin, at least *in vitro*. The evolutionary implications for the presence of cardiolipin in an organism with "bare-bones" mitochondrial remnants *i.e.* mitosomes, with no maintenance of membrane potential nor ATP synthesis activity [66], provide for an exciting research direction worth pursuing.

## An interactome-based model for PIP-binding proteins and clathrin assemblies at PVs

Data derived from APEX-mediated tEM experiments on transgenic trophozoites expressing APEX-tagged clathrin assembly components (*Gl*CHC and *Gl*CLC; [19]) show how larger PVs are associated to more than one PM-derived clathrin-marked invagination (Fig 10A).

This is supported by data from IFA and STED microscopy analysis of trophozoites loaded with Dextran-OG and labelled with anti-*Gl*CHC antibodies (Fig 10B). By combining APEX-derived tEM data with STED microscopy data for both Dextran-OG and *Gl*PXD3 labelling, a quantified sub organellar model for PV organization can be built which takes into account organelle size and relative distribution of clathrin assemblies (Fig 10C). In this model, *Gl*PXD3 clearly emerges as a membrane coat that surrounds individual PV organelles (Fig 10C, upper panel) on the cytoplasmic side of clathrin assemblies at the PV-PM interface (Fig 10C, lower panel).

The PV-associated PIP-binding protein interactome appears as a tightly knit molecular network with *Gl*CHC at its center (Figs 4 and 10D). Despite the high level of interconnectivity of

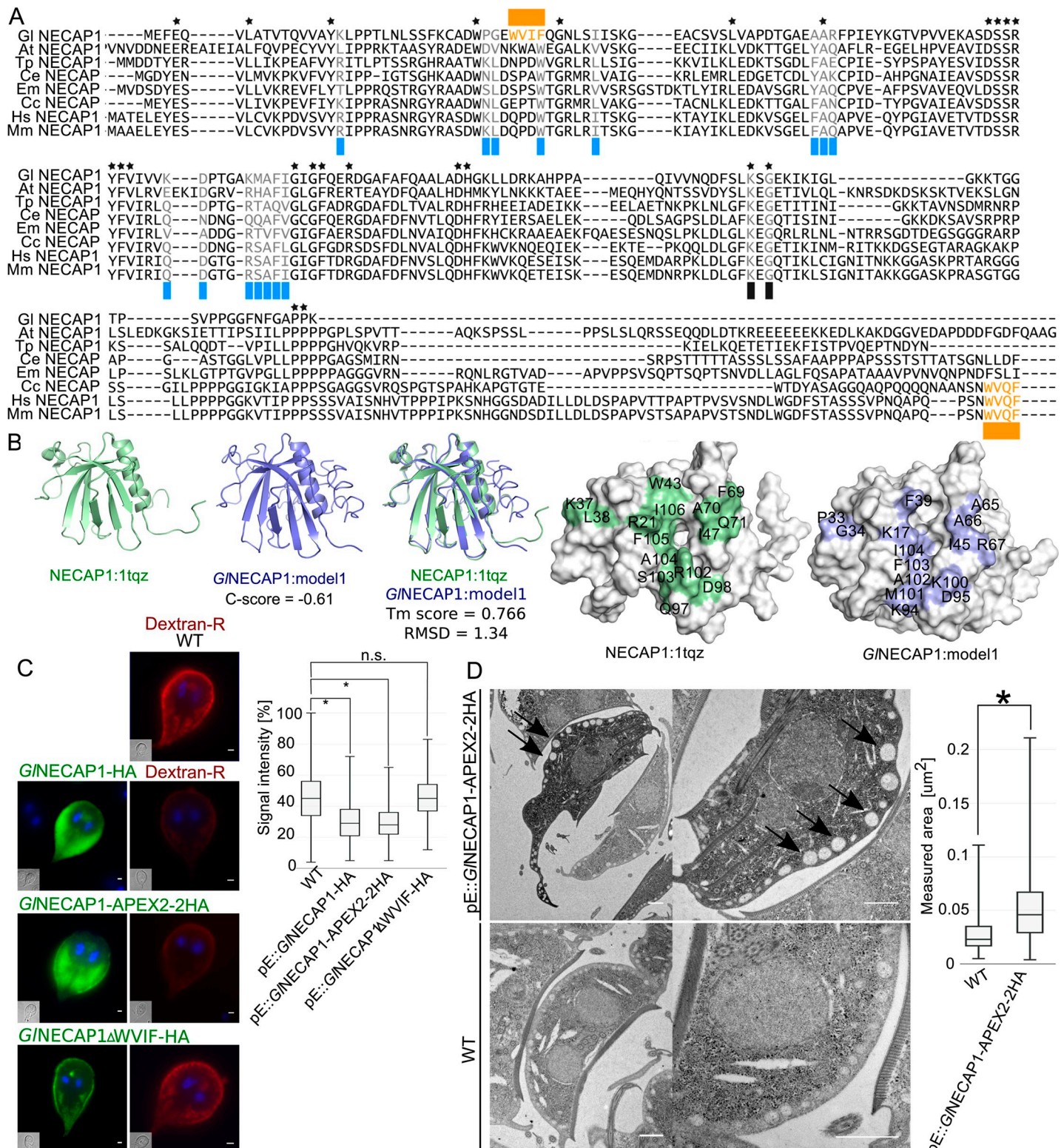

**Fig 8. PV morphology and functionality phenotypes caused by *Gl*NECAP1 ectopic expression.** (A) Multiple sequence alignment analysis of *Gl*NECAP1 and NECAP1 orthologues from *Arabidopsis thaliana* (Uniprot accession Q84WV7), *Trichinella pseudospiralis* (A0A0V1JQ20), *Caenorhabditis elegans* (Q9N489), *Echinococcus multilocularis* (A0A087VZS0), *Ceratitis capitata* (W8CD89), *Homo sapiens* (Q8NC96) and *Mus musculus* (Q9CR95) identifies conserved motifs and residues for interaction with AP2. *Gl*NECAP1 presents partial conservation, with a WXXF motif (orange) shifted to the N-terminus with respect to other orthologues. (B) *Ab initio* template-based 3D modelling of *G. lamblia* and *H. sapiens* NECAP1 (1tqz) homologues predicts similar structures, with conservation of key residues

involved in the interaction between NECAP1 proteins and AP2 complexes (shaded in blue and green). (C) Wild-type non-transgenic control cells (WT) and cells expressing either epitope-tagged *Gl*NECAP1 reporters *Gl*NECAP1-HA, *Gl*NECAP1-APEX2-2HA or the ΔWVIF deletion construct *Gl*NECAP1ΔWVIF-HA (green) were tested for Dextran-R (red) uptake. Dextran-R signal intensity after uptake was significantly (p<0.05) decreased in *Gl*NECAP1-HA- and *Gl*NECAP1-APEX2-2HA-expressing cells when compared to wild-type controls and *Gl*NECAP1ΔWVIF-HA-expressing cells (box-plot). (D) Quantitative tEM analysis of *Gl*NECAP1-APEX2-2HA-expressing cells (upper panels) and wild-type non-transgenic cells (WT; lower panels) shows visibly enlarged PVs in *Gl*NECAP1-APEX2-2HA-expressing cells, with a statistically significant (p<0,05) increase in the median of measured area of peripheral vacuoles (in μm²; box-plot).

distinct PIP-binder interactomes (Fig 4), specific molecular circuits such as the ones defined by the SNARE quartet (S4F Fig), pseudokinase *Gl*15411 and novel DLP *Gl*9605 (S4G Fig), as well as StAR-related lipid-transfer protein *Gl*16717 (S4H Fig), can be recognized. Notably,

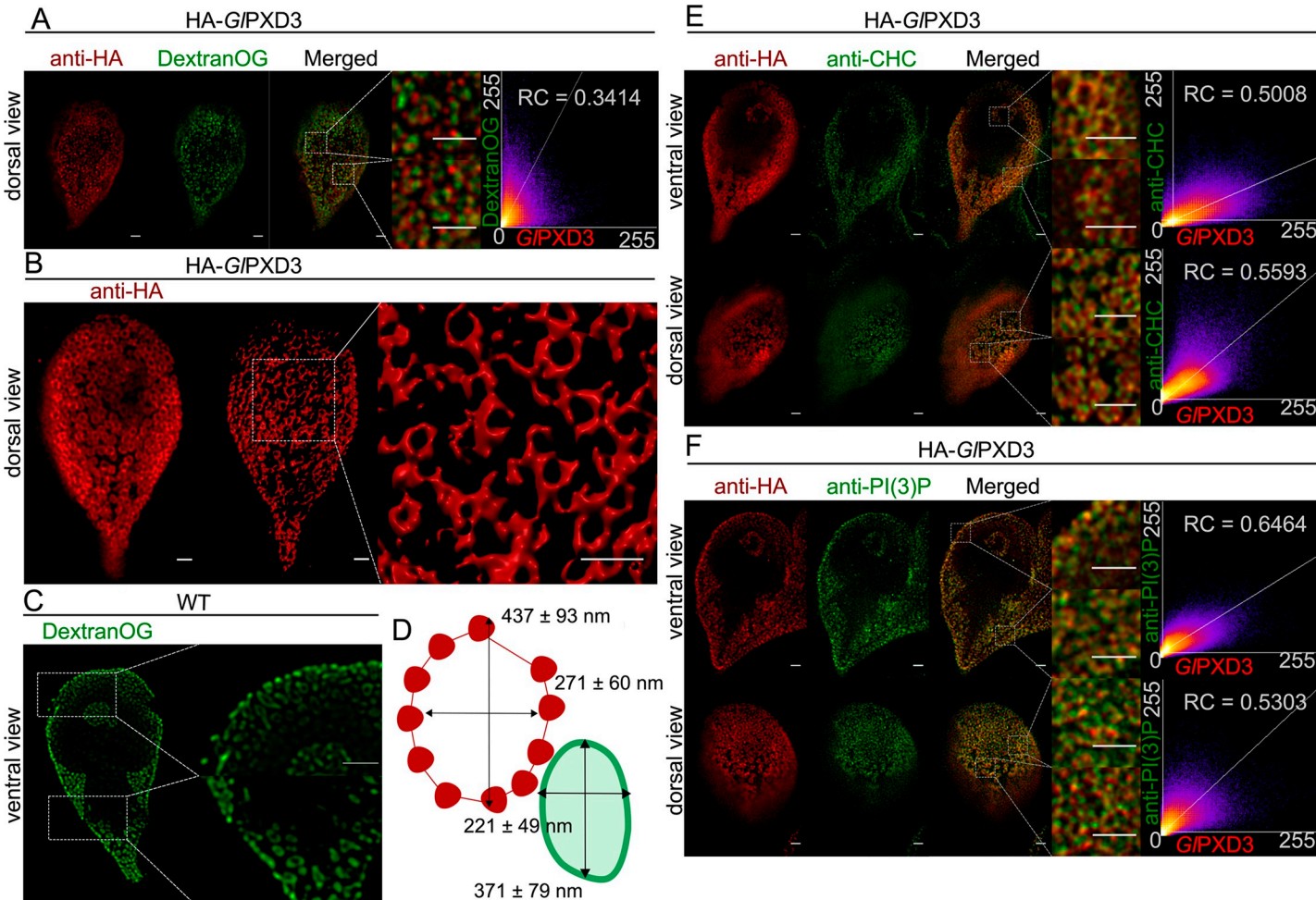

**Fig 9. *Gl*PXD3 membrane coats as a tool to probe PV size and organization.** (A) A dorsal view of representative cells expressing an epitope-tagged *Gl*PXD3 reporter (red) and co-labelled for Dextran-OG (green). STED confocal imaging followed by signal overlap analysis (scatter plot) shows proximal yet distinct deposition patterns, with *Gl*PXD3 reporters closely associated to Dextran-OG-illuminated PVs. Scale bars: whole cell 1 μm; close-ups 1 μm. (B) 3D STED microscopy (left panel) followed by reconstruction using IMARIS (middle panel) of a representative cell expressing an epitope-tagged *Gl*PXD3 reporter reveals fenestrated *Gl*PXD3-delimited areas distributed under the PM and throughout the whole cell (close-up view of inset in the right panel). Scale bars: whole cell 1 μm; close-ups 1 μm. (C) STED microscopy analysis of PVs in a representative non-transgenic wild-type cell labelled with Dextran-OG. Scale bars: whole cell 1 μm; close-ups 1 μm. (D) Average length of the major and minor principle axes of *Gl*PXD3-delimited fenestrated structures (in red) and Dextran-labelled PV organelles in wild-type non-transgenic cells (in green) measured across at least 100 structures/organelles. (E) STED confocal microscopy analysis of ventral and dorsal views of a representative cell expressing an epitope-tagged *Gl*PXD3 reporter (anti-HA) and co-labelled for *Gl*CHC (anti-CHC) shows how fenestrated *Gl*PXD3-delimited structures are decorated with *Gl*CHC foci. Scatter plots are included for signal overlap analysis. Scale bars: whole cell 1 μm; close-ups 1 μm. (F) Similar to *Gl*CHC, anti-PI(3)P antibodies (anti-PI(3)P) detect foci of PI(3)P accumulation in close proximity to *Gl*PXD3 epitope-tagged reporters (anti-HA) in HA-*Gl*PXD3-expressing cells analyzed with STED microscopy. Scatter plots are included for signal overlap analysis. Scale bars: whole cell 1 μm; close-ups 1 μm.

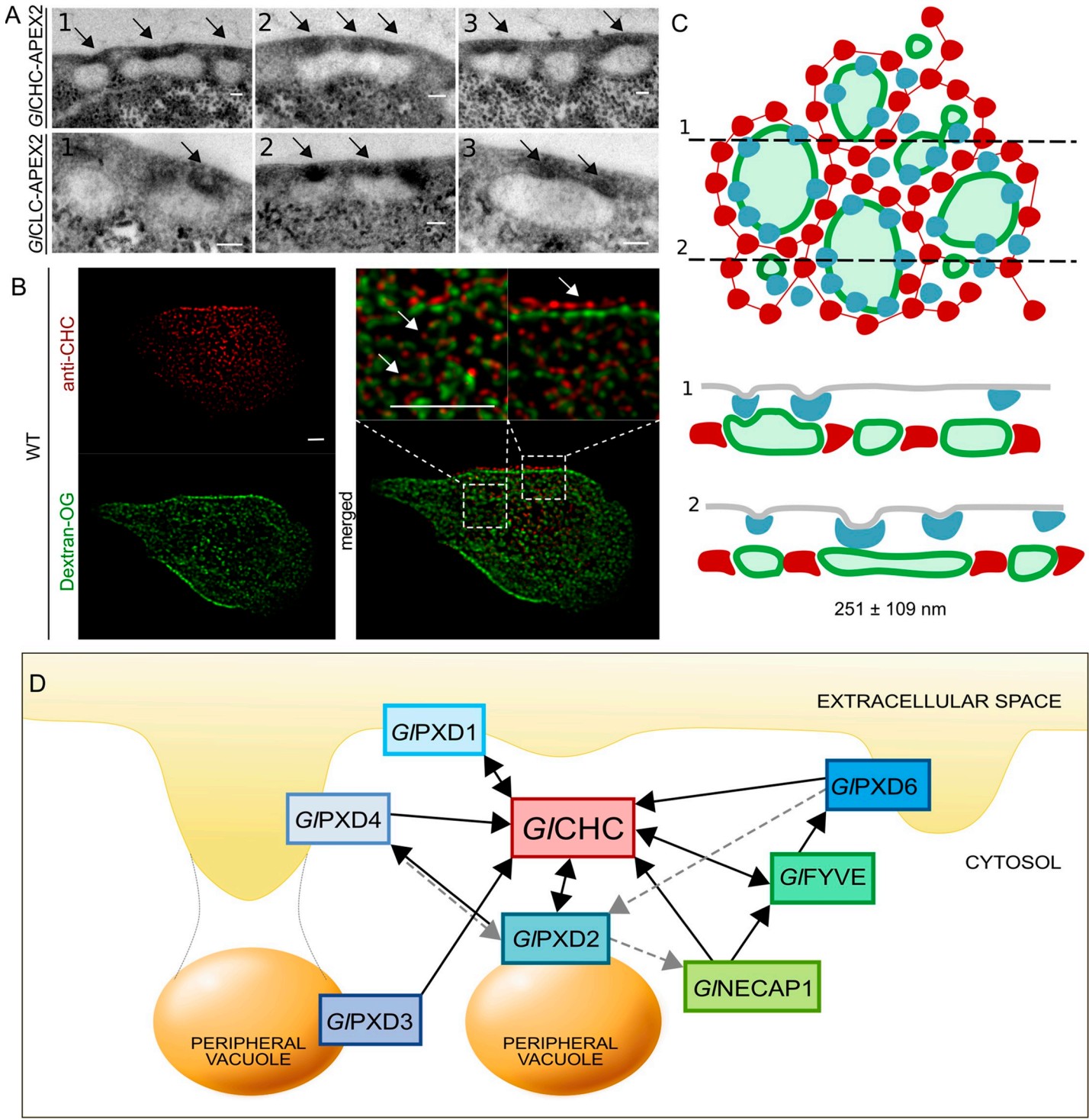

**Fig 10. A working model for PV-associated nanoenvironments defined by clathrin assemblies and PIP-binding proteins.** (A) Electron microscopy images of *G. lamblia* cells expressing an APEX2-tagged *Gl*CHC (upper panels) or *Gl*CLC (lower panels) reporter show darker APEX2-derived deposits at the PM-PV interface (arrows). Scale bar: 0.1 μm. (B) IFA analysis of a representative non-transgenic wild-type cell labelled with Dextran-OG and anti-*Gl*CHC antibodies to illuminate PV lumina and the PV-PM interface, respectively. Scale bar: 1 μm. (C) Schematic reconstruction of a surface view (left panel) of the PV system associated to clathrin assemblies (blue) and *Gl*PXD3 coats (red), based on data presented in this report and in [13]. PV membranes and lumina are represented in dark and light green, respectively. Cross-sections at (1) and (2) yield views in the right panel, highlighting foci of clathrin assemblies beneath the PM, above *Gl*PXD3's coat-like deposition pattern surrounding PVs. (D) An overview of the *G. lamblia* PIP-binding interactome associated to PVs. All represented PIP-binding proteins were found to contact clathrin assemblies (*Gl*CHC) in either reciprocal (double-headed arrows) or one-way (single-headed arrows) modes of interaction, following filtering of co-IP data either at high (black solid lines) or low (grey dashed lines) stringency.

*Gl*PXD1 and 2 are the only PIP-binders who's extended interactomes include the *G. lamblia* putative clathrin light chain (Fig 4 and S4F Fig), arguably *Gl*CHC's closest binding partner. The *Gl*PXD1 interactome further stands out for enrichment of proteasome-associated components (S1 Table), invoking scenarios concerning clathrin assembly turnover in *G. lamblia*. Although previous data showed that clathrin assemblies are long-lived stable complexes [19], they would still require remodeling, degradation, and substitution with new components. In the absence of classical components as well as C-terminal motifs on *Gl*CHC for ordered disassembly of clathrin coats, *Gl*PXD1's proteasome-enriched interactome points to proteasome-mediated degradation of *Gl*CHC assemblies as an alternative process to achieve turnover albeit without recycling of coat components.

In the context of clathrin assembly dynamics, *Gl*NECAP1 once again comes to the forefront. NECAP1 is characterized as an AP2 interacting partner and an important component of CCVs in the assembly phase [57]. Given that CCVs have not been detected in *Giardia*, this begs the question of the functional role of a NECAP1 cardiolipin-binding orthologue in *G. lamblia*, which was found to interact with *G. lamblia* AP2 subunits and *Gl*FYVE. Recent developments in gene knock-out [67] and CRISPR-Cas9-based knock-down [68, 69] methodologies tailored to *G. lamblia* will be instrumental towards a full functional characterization of *Gl*NECAP1's function(s).

## PIP binding homeostasis and fluid-phase uptake

We initially hypothesized that perturbation of free PIP binding sites would elicit fluid-phase uptake phenotypes by impacting PV functionality. The hypothesis tested positive for PI3P, PI(3,4,5)P$_3$, and PI(4,5)P$_2$. A significant effect on cell width was detected when free PI(3)P binding sites were reduced by ectopic expression of 2xFYVE-GFP (Fig 3B and 3F), linking PIP residues to both endocytic homeostasis and overall maintenance of cell size, possibly in connection to membrane turnover. Complementing these data, ectopic expression of both *Gl*FYVE and *Gl*NECAP1 significantly impacted fluid-phase uptake. Furthermore, ectopic expression of *Gl*NECAP1 induced an enlarged PV phenotype similar to that induced by expression of a predicted GTP-locked *Gl*DRP mutant [70]. As with all uptake phenotypes we elicited and measured using fluorescent dextran as a fluid-phase uptake reporter, it is still unclear whether the defect lies in PV-PM fusion or in the sealing-off of PV lumina.

Ectopic expression of a truncated *Gl*FYVE deprived of its ARM repeats, namely NT-*Gl*FYVE, induced formations of vesicle-like structures of undefined origins. ARM folds are superhelical structures mostly involved in protein-protein interactions [71], suggesting that a loss of these domains may impact *Gl*FYVE function and protein complex formation and may lead to protein aggregation. Importantly, these structures are associated to *Gl*CHC which, in cells expressing the NT-*Gl*FYVE recombinant protein, has lost its almost exclusive PV localization. In line with this hypothesis, the NT-*Gl*FYVE epitope-tagged reporter loses association to PVs. In contrast to the *Gl*FYVE-induced uptake phenotype and despite a severe PV clustering phenotype, HA-*Gl*PXD2-expressing cells still appear to perform fluid-phase uptake comparably to wild-type cells. This suggests that PV morphology can be decoupled from effective PV-mediated uptake. Taken together, these data link PIPs to clathrin assemblies and fluid-phase PV-mediated uptake, providing new insights on clathrin's hitherto unclear role in *Giardia* endocytosis.

## Beyond clathrin assemblies

Investigation of the molecular *milieu* within which clathrin-associated PIP-binding proteins operate in *G. lamblia* revealed two protein sets of special interest. Four predicted SNARE

proteins were detected in both the *Gl*PXD2 and *Gl*PXD3 interactomes. Further investigations will be necessary to determine whether the function of this SNARE quartet is indeed fusing PM and PV membranes at contact sites, thereby allowing entry of fluid-phase material into PV organelles.

Another finding of special interest concerns *Gl*9605, a hitherto unrecognized DLP found in the interactome of *Gl*PXD3 with similarity to bacterial DLPs (BDLPs; S6 Fig). Similar to their eukaryotic counterparts, BDLPs are capable of helical self-assembly and tubulation of lipid bilayers, and were shown to be most closely related to the mitofusins FZO and OPA (S6 Fig) [24, 25], but only distantly related to classical dynamins [26]. BDLPs were also found in the Archaea class Methanomicrobia [72], making the family ubiquitously distributed across all kingdoms. These data show how the DLP/DRP family in *G. lamblia* has now expanded to include the previously unidentified endocytosis-associated *Gl*9605 BDLP homologue. *Gl*DRP plays a role in the regulation of PV and encystation-specific vesicle (ESV) size [70]. Although its role in fluid-phase uptake has not been determined, expression of a GTP-locked *Gl*DRP mutant inhibited endocytosis of biotinylated surface proteins [70]. On the other hand, a similar mutational analysis of *Gl*9605 shows that this DLP variant can elicit a dominant-negative fluid-phase uptake phenotype. Although we did not test *Gl*9605 involvement in surface protein uptake, the data so far suggest that two distinct DLPs play independent albeit complementary roles in the regulation of PV-mediated fluid-phase uptake and organelle homeostasis.

In this work, we report on the detailed functional characterization of PIP-binding proteins in *G. lamblia* that associate to clathrin assemblies. Our data reveals a previously unappreciated level of complex interplay between lipid residues and their protein binders in marking and shaping endocytic compartments in this parasite. However, several identified PIP-binding modules appear to associate to PVs independently of clathrin. Their extended interactomes and their involvement in fluid-phase uptake have yet to be investigated but current data point towards a complex network of PIP binders of varying binding preference and affinity, all working in the same subcellular environment, yet, in some cases (*Gl*FERM, *Gl*BAR1 and 2, *Gl*PROP1 and 2, and *Gl*16801), not directly linked to clathrin assemblies. The only known exception is *Gl*epsin whose localization remains controversial due to conflicting reports [21, 73]. We systematically did not detect *Gl*epsin in any of the interactomes for clathrin-associated PIP binders, in line with its localization at the ventral disk [21]. Altogether, the variety of PIP residues and PIP-binding modules in the *G. lamblia* cortical area containing endocytic PVs underscores their necessity for correct functioning of membrane traffic even in a protist so clearly marked by reduction in endomembrane complexity.

## Materials and methods

### *Giardia lamblia* cell culture, induction of encystation and transfection

*G. lamblia* WBC6 (ATCC catalog number 50803) trophozoites were cultured and harvested applying standardized protocols [56]. Encystation was induced by the two-step method as previously described [74, 75]. Transgenic cell lines were generated using established protocols by electroporation of linearized or circular pPacV-Integ-based plasmid vectors prepared from *E. coli* as described in [76]. Transgenic lines were then selected for puromycin resistance (final conc. 50 µg ml $^{-1}$). After selection, transgenic trophozoites carrying episomal or integrated reporter constructs were further cultured with or without puromycin, respectively.

### Construction of expression vectors

Oligonucleotide sequences used for cloning in this work are listed in S8 Table. pPacV-Integ-based [34] expression of epitope tagged reporter constructs was driven using either putative

endogenous (pE) or encystation-dependent (pCWP1) promoters. Constructs 2xFYVE-GFP and GFP-P4C [42] were kindly provided by Prof. Dr. H. Hilbi (University of Zurich).

## PV labelling using fluid-phase markers

Fluid-phase uptake assay in *G. lamblia* was performed as described previously [26] using dextran coupled to either Oregon Green 488 (Dextran-OG) (Cat. Nr. D-7171, Thermo Fisher Scientific) or Texas-Red (Dextran-R) (Cat. Nr. D-1863, Thermo Fisher Scientific) fluorophores, both at 1mg/ml final concentration. The same protocol was used following treatment with 7.2mM neomycin (G418, Sigma) in supplemented PBS for 45–50 minutes at 37˚C.

## Co-immunoprecipitation with limited cross-linking

Co-immunoprecipitation of *Gl*PXD1-6, *Gl*NECAP1, and *Gl*FYVE was done as previously reported [19, 45]. Protein input was standardized to 0.8 mg/ml total protein.

## Protein analysis and sample preparation for mass spectrometry (MS)-based protein identification

Protein analysis was performed on 4%/10% polyacrylamide gels under reducing conditions (molecular weight marker Cat. Nr. 26616, Thermo Scientific, Lithuania). Immunoblotting was done as described in [77]. Gels for mass spectrometry (MS) analysis were stained using Instant Blue (Expedeon, Prod. # iSB1L) and destained with ultra-pure water.

## Mass Spectrometry, protein identification and data storage

MS-based protein identification of de-stained and diced gel lanes was performed as described in [19]. Free access to raw MS data is provided through the ProteomeXchange Consortium on the PRIDE platform [78]. Accession numbers for datasets derived from bait-specific and corresponding control co-IP MS analyses are the following: PXD013890 for *Gl*PXD1, 3 and 6, PXD013897 for *Gl*FYVE, PXD013896 for *Gl*NECAP1 and PXD013899 for *Gl*PXD2 and 4.

## *In silico* co-immunoprecipitation dataset analysis

Analysis of primary structure and domain architecture of putative components of giardial PIP—binding proteins was performed using the following online tools and databases: SMART for prediction of patterns and functional domains (http://smart.embl-heidelberg.de/), pBLAST for protein homology detection (https://blast.ncbi.nlm.nih.gov/Blast.cgi?PAGE=Proteins), HHPRED for protein homology detection based on Hidden Markov Model (HMM-HMM) comparison (https://toolkit.tuebingen.mpg.de/#/tools/hhpred), PSORTII for sub-cellular localization prediction (https://psort.hgc.jp/form2.html), TMHMM for transmembrane helix prediction (http://www.cbs.dtu.dk/services/TMHMM/), RCSB for 3D structure of homologues (https://www.rcsb.org/), and the Giardia Genome Database to extract organism-specific information such as protein expression levels, predicted molecular sizes and nucleotide/protein sequences (www.giardiaDB.org). The generated co-IP datasets were filtered using a dedicated control-co-IP dataset generated using non-transgenic wild-type parasites. Filtration of the bait-specific co-IP and control-co-IP datasets was done using Scaffold4 (http://www.proteomesoftware.com/products/) with high stringency parameters (95_2_95, FDR 0%) and low stringency parameters (95_2_50, FDR 0%). Furthermore, exclusive hits for bait-specific datasets were manually curated using the following criteria for inclusion into the interactome model: i) exclusive detection with > 3 spectral counts in bait-specific datasets or ii) an

enrichment of peptide counts >3 with respect to the ctrl. co-IP dataset. Data presented in S1– S7 Tables show exclusive and non-exclusive protein hits filtered using both stringency levels.

## Immunofluorescence analysis (IFA) and light-microscopy

Samples for immunofluorescence analysis of subcellular distribution of reporter proteins by wide-field and laser scanning confocal microscopy (LSCM) were prepared as described previously [33, 35]. Nuclear DNA was labelled with 4', 6-diamidino-2-phenylindole (DAPI). Antibodies used are the anti-HA antibody (1:50 or 1:100; Anti HA high affinity 3F10, Cat. Nr. 11867423001, Roche), anti-V5 (1:50 or 1:100; V5 Tag Monoclonal Antibody, Cat. Nr. R960-25, Thermo Fisher Scientific), self-made antibodies raised against *Gl*CHC (dilution 1:1000), anti-rat antibody coupled to fluorochrome in case of wide-field or confocal microscopy (1:200; Goat anti-Rat IgG (H+L) Cross-Adsorbed Secondary Antibody, Alexa Fluor 488, Cat. Nr. #A11006, Invitrogen) and for STED microscopy (Goat anti-Rat IgG (H+L) Cross-Adsorbed Secondary Antibody, Alexa Fluor 594, Cat. Nr. A11007, Invitrogen). Specific PIP residues were detected using anti-PI(3)P (1:100; Purified anti-PI(3)P IgG, Z-P003 Echelon Biosciencies), anti-PI(4,5)$P_2$ (1:100; Purified anti-PI(4,5)$P_2$ IgM, Z-P003 Echelon Biosciencies) and anti-PI(3,4,5)$P_3$ (1:100; Purified anti-PI(3,4,5)$P_3$ IgM, Z-P045 Echelon Biosciencies) followed by an anti-mouse antibody coupled to fluorochrome in all three cases (Goat anti-Mouse IgG (H+L) Cross-Adsorbed Secondary Antibody, Alexa flour 594, Cat. Nr. A-11005, Thermo Fischer Scientific or Goat anti-Mouse IgG (H+L) Cross-Adsorbed Secondary Antibody, Alexa flour 488, Cat. Nr. A-11017, Thermo Fischer Scientific). Cells were imaged at maximum width, with nuclei and the bare-zone at maximum diameter. Deconvolution was performed with Huygens Professional (Scientific Volume Imaging). Three-dimensional reconstructions and signal overlap quantification (Mander's coefficient) in volume images of reconstructed stacks were performed using IMARIS x64 version 7.7.2 software suite (Bitplane AG) or FIJI [79], respectively.

## Live cell microscopy of GFP fusion proteins in transgenic *Giardia lamblia*

Transgenic trophozoites expressing GFP-fusion proteins and non-transgenic lines were harvested and resuspended in approximately 0.5–1 ml of the medium and transferred to a precooled 24 well culture plate placed directly on ice in an ice bucket. After overnight oxygenation in the dark at 4˚C, cells were washed in ice-cold PBS supplemented (S-PBS) with 5mM glucose (Cat. No. 49139, Fluka) and 0.1mM ascorbic acid (Cat. No. 95209, Fluka) at pH 7.1. An aliquot of cells in S-PBS was placed on a microscopy slide and left to recover at 37˚C for 3 minutes directly before imaging. For the Dextran uptake assay, Dextran-R was added to the cell suspension to a final concentration of 1mg/ml. Cells were incubated in the dark at 37˚C for 20 min and were imaged directly or chemically fixed for further processing by IFA.

## Super resolution (gSTED) microscopy

Sample preparation was done as described for wide field microscopy and LSCM. For imaging, samples were mounted in ProLong Gold antifade reagent (Cat. Nr. P36934, Thermo Fisher Scientific). Super resolution microscopy was performed on a LSCM SP8 gSTED 3x Leica (Leica Microsystems) at the Center for Microscopy and Image Analysis, University of Zurich, Switzerland. Nuclear labelling was omitted due to possible interference with the STED laser. Further data processing and three-dimensional reconstructions of image stacks were done as described for LSCM.

## Sample preparation for transmission electron microscopy

Transgenic trophozoites expressing *Gl*PXD2 (GL50803_16595) and non-transgenic cells were harvested and analyzed by transmission electron microscopy (tEM) as described previously [70].

## DAB staining in APEX2 expressing cells

Transgenic trophozoites expressing *Gl*NECAP1-APEX2-2HA, *Gl*CHC-APEX2-2HA and *Gl*CLC-APEX2-2HA were harvested and washed with PBS followed by fixation in 2.5% EM grade glutaraldehyde in cacodylate buffer (100 mM cacodylate (Cat. Nr. 20838), 2mM CaCl$_2$ (Cat. Nr. 21097, Fluka) in PBS) for 1h at RT. Samples were washed twice before and after quenching for 5 min in 20 mM glycine/cacodylate buffer. For staining, cells were resuspended in 500 μl substrate solution containing 1.4 mM DAB tetrahydrochloride (Cat. Nr. D5637, Sigma) with 0.3 mM H$_2$O$_2$ (Cat. Nr. H1009, Sigma) in cacodylate buffer and incubated for 15 min. The reaction was terminated by washing thrice in cacodylate buffer and prepared as described for tEM.

## Chemical fixation of DAB-stained cells

DAB stained cell suspicions were post-fixed with 1% aqueous OsO4 for 1 hour on ice, subsequently rinsed three rimes with pure water and dehydrated in a sequence of ethanol solutions (70% up to 100%), followed by incubation in 100% propylene oxide and embedding in Epon/Araldite (Sigma-Aldrich, Buchs, Switzerland). Samples were polymerized at 60˚C for 24h. Thin sections were imaged pre- and post- staining with aqueous uranyl acetate (2%) and Reynolds lead citrate.

## Expression and purification of bacterial fusion proteins

For each candidate PIP-binding protein, corresponding nucleotide stretches coding for selected amino acid residues (S9 Table) were modified by including an HA-coding sequence either at the 5' end or the 3' end and then subcloned into the pMal-2Cx *E. coli* expression vector (New England Biolabs). The resulting recombinant variants were expressed as maltose-binding protein (MBP) fusions in *E. coli* (strain Bl21) and grown in LB medium either at 37˚C (MBP-*Gl*PXD1, MBP-*Gl*PXD2, MBP-*Gl*PXD3, MBP-*Gl*PXD6, MBP-*Gl*NECAP1 and MBP-*Gl*FYVE) or 30˚C (MBP-*Gl*PXD4 and MBP-*Gl*PXD5) to an OD$_{600}$ = 0.4. Induction of expression was performed by adding 0.2 mM IPTG (Isopropyl β-D-1-thiogalactopyranoside, Cat. Nr. 15529019, Thermo Fischer Scientific) to the cultures and incubating for a further 4 hours. Cells were harvested at 4˚C (4,000 x g) and bacterial pellets were resuspended in 5 ml of cold column buffer with 1x PIC (Protease inhibitor cocktail set I; Cat. Nr. 539131-10VL, Merck) and 200 mM PMSF (Cat. Nr. 329-98-6, Sigma Aldrich). Cells were lysed by sonication and centrifuged (20 min, 9,000 x g, 4˚C). Cleared supernatant was incubated with amylose resin slurry (Amylose Resin High Flow, Cat. Nr. E8022L, BioLabs) for 4 hours at 4˚C on a turning wheel, washed with column buffer and then transferred to an empty column (BioRad). Unbound protein was washed using until background OD$_{280}$ reached ~0.06. Protein fractions were eluted using 10mM maltose solution and pooled for overnight dialysis in a dialysis cassette (Slide-a-Lyzer, Cat. Nr. 66380, Thermo Fischer Scientific) against 25mM NH$_4$Ac at 4˚C and later lyophilized or snap-frozen. Protein fractions were stored at -80˚C.

## Protein lipid overlay (PLO) assay

PLO assays were performed using two different types of PIP strips (Cat. Nr. P-6001 and P-6002, Echelon) and PIP arrays (Cat. Nr. P-6100, Echelon). PIP strips or PIP arrays were first

floated on ultrapure water for 5 min before incubation in blocking buffer (1xPBS containing 0.1%v/v Tween-20 and 3% fatty-acid free BSA (Sigma A7030)) at RT for 1h. For PLO using lipid strips, *E. coli*-derived lyophilized were reconstituted in 1x PBS and protein concentration was measured using the Bradford assay. For PLO using PIP arrays, snap-frozen protein samples were used. For both types of PLO assays, the equivalent of 0.5 μg/ml of protein in PBS containing 3% fatty acid free BSA were incubated for 1h at RT with gentle agitation. After washing with 1xPBS containing 0.1% v/v Tween-20, PIP–strips were incubated (1h, RT, agitated) with a monoclonal anti-HA antibody (clone 3F10, monoclonal antibody from Roche) at a dilution of 1:500 in blocking buffer. Subsequently strips were washed and incubated (1h, RT, agitated) with a goat-derived polyclonal anti-rat antibody conjugated to HRP at a dilution of 1:2000 in blocking buffer (Cat. Nr. 3050–05, Southern Biotech). After further washing, strips were developed using a chemiluminescent substrate (WesternBright ECL HRP Substrate, Cat. Nr. K-12045-D50).

## Densitometric analysis of lipid strips and arrays

Relative quantification of immunoblotting signal intensity on PIP strips and arrays overlaid with PIP-binding proteins was performed using FIJI [79]. For each strip or array, the spot with the highest pixel number was set as a reference for 100% binding; signals coming from all other spots were normalized against it. The data were visualized as bar charts of relative signal intensity as a measure of lipid-binding preference for each PIP-module.

## Identification of *Giardia* orthologues of known PIP-binding domains

PIP-binding domain representatives were used as bait for *in silico* searches within the *Giardia* genome database (GDB) (http://giardiadb.org/) using the online tool HHpred (https://toolkit.tuebingen.mpg.de/) to detect remote giardial homologues using hidden Markov models (HMMs; Table 1) [25]. Outputs were firstly evaluated based on the calculated probability and the corresponding E-value for the prediction, with cut-offs for probability and e-value set to 90 and 1e-10, respectively. Sequence identity and similarity were also considered. To validate the prediction, candidate giardial PIP-binding proteins were then utilized as baits to search PDB databases using HHpred to retrieve orthologous PIP-binding proteins/modules. For additional validation, I-TASSER [32–34] was also used to predict hypothetical structures of putative giardial PIP-binding domains next step validation.

## Multiple sequence alignment analysis

Multiple sequence alignment using two or more sequences was performed with the Clustal Omega sequence alignment algorithm [80, 81]. The sequences used to compile the alignments shown in S1 Fig were chosen based on representative members for each PIP-binding domain type [1, 10, 82]. Alignments for Figs 6 and 8 were based on previously characterized G1-G4 GTP binding motifs [53] and NECAP1 proteins [57], respectively.

## *De novo* structural modeling and analysis

*Ab-initio* prediction of hypothetical 3D models presented in S1 Fig was done using I-TASSER [32–34]. The best model was chosen based on the C-score predicted by the algorithm. A C-score is a measure of confidence for a model based on the significance of threading template alignments and the convergence parameters of the structure assembly simulations. It ranges from -5- to 2, with higher C-scores indicating higher confidence. The final 3D structures were displayed using PyMOL (The PyMOL Molecular Graphics System, Version 2.0 Schrödinger,

**Table 3. GTPase domain sequences selected for the phylogenetic analysis of *Gl*9605, a novel *Giardia* BDLP.** For each entry, the origin (species and abbreviation), assigned name (protein), unique identifier (UniProtKB) and amino acid stretch used to reconstruct the tree shown in S6 Fig, are provided.

| Species and abbreviation | Protein | UniProtKB Identifier | Residue numbers |
|---|---|---|---|
| *Aa—Aquifex aeolicus* | Aa FtsZ | O66809 | 1–240 |
| *Bv—Bacillus velezensis* | Bv DynA | S6FLE6 | 1–240 |
| *Rn—Rattus norvegicus* | Rn MFN1 | Q8R4Z9 | 1–240 |
| *Hs—Homo sapiens* | Hs MFN1 | Q8IWA4 | 1–240 |
| | Hs Dyn1 | Q05193 | 1–240 |
| | Hs Dyn3 | Q9UQ16 | 1–240 |
| | Hs Dyn2 | P50570 | 1–240 |
| | Hs MFN2 | O95140 | 1–240 |
| | Hs DNM1L | O00429 | 1–240 |
| | Hs ATL1 | Q8WXF7 | 1–240 |
| | Hs GBP1 | P32455 | 1–240 |
| | Hs GBP5 | Q96PP8 | 1–240 |
| | Hs GBP3 | Q9H0R5 | 1–240 |
| | Hs GBP2 | P32456 | 1–240 |
| | Hs MX1 | P20591 | 1–240 |
| | Hs OPA1 | O60313 | 1–277 |
| | Hs ATL3 | Q6DD88 | 1–240 |
| | Hs MX2 | P20592 | 1–240 |
| | Hs GBP4 | Q96PP9 | 1–240 |
| | Hs ATL2 | Q8NHH9 | 1–246 |
| *Np—Nostoc punctiforme* | Np BDLP | B2IZD3 | 1–240 |
| *Dd—Dictyostelium discoideum* | Dd DynA | Q94464 | 1–240 |
| *Ss—Synechocystis sp.* | Ss FtsZ | P73456 | 1–240 |
| *Ec—Escherichia coli* | Ec LeoA | Q9RFR9 | 1–240 |
| | Ec FtsZ | AJF44969.1 | 1–280 |
| *Mm—Mus musculus* | Mm ATL1 | Q8BH66 | 1–240 |
| *Sc—Saccharomyces cerevisiae* | Sc SEY1 | Q99287 | 1–240 |
| | Sc FZO1 | P38297 | 1–240 |
| *Ce—Caenorhabditis elegans* | Ce FZO1 | Q23424 | 1–240 |
| *Gl—Giardia lamblia* | Gl DRP | E2RU04 | 1–320 |
| | Gl 9605 | A8BAT2 | 1–320 |
| *Bc—Bacillus cereus* | Bc BDLP | CUB17917.1 | 1–280 |
| *A—Anoxybacillus sp.* | A BDLP1 | KXG09432.1 | 1–280 |
| *Cr -Chlamydomonas reinhardtii* | Cr FtsZ | BAB91150.1 | 1–280 |
| *Pa—Pseudomonas aeruginosa* | Pa BDLP | AMT98798.1 | 1–280 |
| *At—Agrobacterium tumefaciens* | At FtsZ | AAC45821.1 | 1–280 |
| *Sm—Sinorhizobium meliloti* | Sm FtsZ | AAC45824.1 | 1–280 |
| *Lysinibacillus saudimassiliensis* | Ls BDLP | CEA00228.1 | 1–280 |

LLC.). The superimposition of *Giardia* PIP-binding proteins with their closest structural orthologue are based on I-TASSER predictions, with structural similarities expressed by TM-score and RMSD[a] values. The TM-score is computed based on the C-score and protein length. It ranges from 0 to 1, where 1 indicates a perfect match between two structures. RMSD[a] is the root mean square deviation between residues that are structurally aligned by TM-align [83]. Specifically for *Gl*BAR1 and 2, the structural overlap analysis was performed by selecting positively-charged residues from previously characterized BAR domains shown to play a role in

lipid binding [84]. These were manually superimposed on corresponding residues in the predicted *Gl*BAR1 and 2 structures.

## Phylogenetic analysis

Selected sequences of GTPase domains (Table 3) were aligned using Clustal Omega tool. The tree construction was submitted to a PHYLogeny Inference Package (PHYLIP) program [85, 86] using random number generator seed set to 111 and number of bootstrap trials set to 10000. The tree was visualized using the on-line tool iTOL and includes branch lengths as a measure of evolutionary distance [87].

## Supporting information

**S1 Fig. Multiple sequence alignment and structural prediction analysis of *G. lamblia* PIP-binding domains.** For all PIP-binding modules except *Gl*NECAP1, the sequence of the lipid-binding domain was aligned to its respective homologous domains. Each domain was structurally modelled using I-TASSER (blue) and superimposed on its closest structural homolog (green). For each structural overlay a TM-score and RMSD value are reported, followed by a blow-up of the calculated location of known and predicted PIP-binding motifs. (A-B) *Gl*epsin, (C-E) *Gl*FYVE and *Gl*16801, (F-L) *Gl*PXD1-6, (M-O) *Gl*BAR1 and *Gl*BAR2, (P-Q) *Gl*FERM and (R-T) *Gl*PROP1 and *Gl*PROP2. (U) Legend to color code for conserved/similar residues. (V) Closest structural homologues, including their origin and identifiers, for structural overlay analysis of PIP-binding modules in *G. lamblia*. (W) Selected orthologues, including their origin and identifiers, for each *G. lamblia* PIP-binding module used in the MSA analysis to highlight conserved/similar residues for lipid-binding.
(TIF)

**S2 Fig. Lipid-binding properties of *Giardia*-lipid binding domains.** Lipid-binding and immuno-detection analysis of *G. lamblia* PIP-binding domains from proteins *Gl*PXD1-6, *Gl*FYVE and *Gl*NECAP1 using lipid strips. (A) Schematic diagram of the pMAL-p2Cx vector used for heterologous expression of individual PIP-binding domains in *E.coli*. (B) SDS-PAGE analysis of recombinant epitope-tagged MBP-PIP binding domain fusions normalized to 1μg total protein. Protein ladder sizes are included in the first lane. (C) Immuno-detection of epitope-tagged MBP-fusions for each PIP-binding domain overlaid on lipid strips carrying spotted lipid residues at 100 pmol/spot and visualized by chemiluminescence. (D) Lipid binding preferences for all tested MBP-domain fusions, measured using FIJI and visualized as plots of relative signal intensity for each probed lipid residue. Values were normalized to those of lipid residues presenting strongest signal intensity. (E) Lipid binding preferences for *Gl*NECAP1 and *Gl*FYVE investigated using a different set of spotted lipid residues revealed *Gl*NECAP1's exclusive affinity for cardiolipin.
(TIF)

**S3 Fig. Subcellular distribution of PI(3)P, PI(4,5)P$_2$ and PI(3,4,5)P$_3$ in *G. lamblia* trophozoites.** 3D STED microscopy analysis followed by signal overlap and deconvolution of representative non-transgenic wild-type *G. lamblia* trophozoites co-labeled with anti-*Gl*CHC (in green) antibody and either (A-B) anti-PI(3)P, (C-D) anti-PI(4,5)P$_2$, or (E-F) anti-PI(3,4,5)P$_3$ antibodies (in red). Dorsal and ventral sides are defined with respect to the ventral disk. Scale bar for (A, C, E): 1 μm. Scale bar for (B, D, F): 2 μm. Scale bar for insets in (B, D, F): 0.5 μm.
(TIF)

**S4 Fig. Selected PIP-binding protein sub-interactomes and PV-associated nano-environments.** (A) Curated interactomes for *Gl*PXD1, *Gl*PXD4 and *Gl*PXD6. All three epitope-tagged

variants used as affinity handles in co-immunoprecipitation experiments identify *Gl*CHC as a strong interaction partner for *Gl*PXD1, 4 and 6. *Gl*PXD1 and 4 further interact with other known clathrin assembly components such as *Gl*CLC, *Gl*AP2 subunits α, β and μ, and *Gl*DRP. *Gl*PXD2, albeit at low stringency, is the only other PXD protein found in all three interactomes. The *Gl*PXD4 interactome includes a putative SNARE protein (5785; [45]) while *Gl*PXD6 as an affinity handle pulled down another PIP residue binder, *Gl*FYVE, known to be associated to clathrin assemblies in *G. lamblia* [19]. (B) The curated extended interactome for *Gl*PXD2 includes all core clathrin assembly components (*Gl*CHC, *Gl*CLC, all *Gl*AP2 subunits, *Gl*DRP [19]) and includes PIP-binders *Gl*PXD4 and *Gl*NECAP1. Three putative SNAREs *Gl*5785, *Gl*10013 and *Gl*14469 were also detected as *bona fide* *Gl*PXD2 interaction partners, the latter previously detected in the *Gl*PXD4 interactome. (C) Analysis of the extended *Gl*PXD3 interactome using an epitope-tagged variant as affinity handle reveals robust interactions with clathrin assembly components *Gl*CHC, α and β *Gl*AP2 subunits, and *Gl*DRP. Predicted inactive NEK kinase 15411 [48] is similarly associated to clathrin assemblies [19] and further shares proteins *Gl*16811, *Gl*87677 and *Gl*17060 as interaction partners with *Gl*PXD3. Predicted SNARE protein *Gl*7309, *Gl*NSF (GL50803_1154776) and proteins *Gl*103709 and *Gl*9605 are unique *Gl*PXD3 interaction partners. The *Gl*PXD3 interactome is connected to the *Gl*PXD6 circuit both directly and through *Gl*16717. (D) The extended interactome analysis of epitope-tagged *Gl*FYVE confirmed tight association to *Gl*CHC, *Gl*DRP and *Gl*PXD6. *Gl*NECAP1 as an alternative PIP-binding module was also detected. (E) A *Gl*NECAP1-centered interactome highlights association to clathrin assembly components and to additional PIP-residue binders *Gl*FYVE and *Gl*PXD2. For all interactomes: solid lines: interactions detected at high stringency; dashed lines: interactions detected at low stringency; yellow protein interaction partners are currently annotated on giardiaDB.org as "hypothetical" *i.e.* of unknown function. (F-H) Nanoenvironments defined by specific sets of interaction partners including clathrin assemblies, PIP-binding proteins, SNARES and proteins of currently unknown function.
(TIF)

**S5 Fig. APEX-mediated electron microscopy analysis of *Gl*NECAP1 subcellular deposition.** (A-F) Representative images of transgenic *G. lamblia* cells expressing construct pE::*Gl*NECAP1-APEX2-2HA showing enlarged PVs and a diffused APEX-dependent cell staining signal. (G-I) Non-transgenic control cells. Scale bar: 1μm.
(TIF)

**S6 Fig. Phylogenetic analysis and tree reconstruction for the predicted GTPase domain of the novel dynamin-like protein *Gl*9605.** Phylogenetic analysis of predicted GTPase domains from the following prokaryotic and eukaryotic species used to compute the tree shown in the figure, including branch lengths as a measure of evolutionary distance: Aa—*Aquifex aeolicus*, Bv—*Bacillus velezensis*, Rn—*Rattus norvegicus*, Hm—*Homo sapiens*, Np—*Nostoc punctiforme*, Dd—*Dictyostelium discoideum*, Ss—*Synechocystis sp.*, Ec—*Escherichia coli*, Mm—*Mus musculus*, Sc—*Saccharomyces cerevisiae*, Ce—*Caenorhabditis elegans*, Gl—*Giardia lamblia*, Bc—*Bacillus cereus*, Ls—*Lysinibacillus saudimassiliensis*, A—*Anoxybacillus sp.*, Cr—*Chlamydomonas reinhardtii*, Pa—*Pseudomonas aeruginosa*, At—*Agrobacterium tumefacies*, Sm—*Sinorhizobium meliloti*.
(TIF)

**S1 Table. Proteins identified in the interactome of *Gl*PXD1.**
(XLSX)

**S2 Table. Proteins identified in the interactome of *Gl*PXD2.**
(XLSX)

**S3 Table. Proteins identified in the interactome of *Gl*PXD3.**
(XLSX)

**S4 Table. Proteins identified in the interactome of *Gl*PXD4.**
(XLSX)

**S5 Table. Proteins identified in the interactome of *Gl*PXD6.**
(XLSX)

**S6 Table. Proteins identified in the interactome of *Gl*FYVE.**
(XLSX)

**S7 Table. Proteins identified in the interactome of *Gl*NECAP1.**
(XLSX)

**S8 Table. List of oligonucleotide names and sequences for construct synthesis.**
(XLSX)

**S9 Table. Amino acid sequences of lipid-binding modules used *in vitro* for protein lipid-overlay assay.**
(XLSX)

**S1 Video. Motility of non-transgenic WB cells following treatment for 50 minutes at 37˚C with 0mM neomycin.**
(M4V)

**S2 Video. Motility of non-transgenic WB cells following treatment for 50 minutes at 37˚C with 2mM neomycin.**
(M4V)

**S3 Video. Motility of non-transgenic WB cells following treatment for 50 minutes at 37˚C with 7.2mM neomycin.**
(M4V)

**S4 Video. Motility of non-transgenic WB cells following treatment for 50 minutes at 37˚C with 15mM neomycin.**
(M4V)

## Acknowledgments

Florian Schmidt and Ritta Rabbat are acknowledged for their technical assistance. Dr. Jon Paulin Zumthor is gratefully acknowledged for his assistance in the early stages of this work. Dr Andres Kaech and the Center for Microscopy and Image Analysis (University of Zurich) are gratefully acknowledged for support and guidance on both light and electron microscopy experiments. Dr Lars Malmstroem and Dr Hamed Khakzad (University of Zurich) are gratefully acknowledged for support and guidance on *ab initio* structural modelling approaches. Prof. Dr. Hubert Hilbi (University of Zurich) is gratefully acknowledged for sharing *Legionella*-derived constructs 2xFYVE-GFP and GFP-P4C.

## Author Contributions

**Conceptualization:** Lenka Cernikova, Carmen Faso, Adrian B. Hehl.

**Data curation:** Lenka Cernikova, Carmen Faso.

**Formal analysis:** Lenka Cernikova, Carmen Faso.

**Funding acquisition:** Carmen Faso, Adrian B. Hehl.

**Investigation:** Lenka Cernikova, Carmen Faso.

**Methodology:** Lenka Cernikova, Carmen Faso, Adrian B. Hehl.

**Project administration:** Carmen Faso, Adrian B. Hehl.

**Resources:** Adrian B. Hehl.

**Supervision:** Carmen Faso, Adrian B. Hehl.

**Validation:** Lenka Cernikova, Carmen Faso.

**Visualization:** Lenka Cernikova.

**Writing – original draft:** Lenka Cernikova, Carmen Faso, Adrian B. Hehl.

**Writing – review & editing:** Lenka Cernikova, Carmen Faso, Adrian B. Hehl.

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
