## [Decision Letter · Decision Letter 0]

7 Oct 2019

Dear Prof. Hehl,

Thank you very much for submitting your manuscript "Phosphoinositide-binding proteins mark, shape and functionally modulate highly-diverged endocytic compartments in the parasitic protist Giardia lamblia" (PPATHOGENS-D-19-01613) for review by PLOS Pathogens. Your manuscript was fully evaluated at the editorial level and by independent peer reviewers. The reviewers appreciated the attention to an important problem, but raised some substantial concerns about the manuscript as it currently stands. These issues must be addressed before we would be willing to consider a revised version of your study. We cannot, of course, promise publication at that time.

We therefore ask you to modify the manuscript according to the review recommendations before we can consider your manuscript for acceptance. Your revisions should address the specific points made by each reviewer.

(1) A letter containing a detailed list of your responses to the review comments and a description of the changes you have made in the manuscript. Please note while forming your response, if your article is accepted, you may have the opportunity to make the peer review history publicly available. The record will include editor decision letters (with reviews) and your responses to reviewer comments. If eligible, we will contact you to opt in or out.

(2) Two versions of the manuscript: one with either highlights or tracked changes denoting where the text has been changed; the other a clean version (uploaded as the manuscript file).

Additionally, to enhance the reproducibility of your results, PLOS recommends that you deposit your laboratory protocols in protocols.io, where a protocol can be assigned its own identifier (DOI) such that it can be cited independently in the future. For instructions see http://journals.plos.org/plospathogens/s/submission-guidelines#loc-materials-and-methods

We hope to receive your revised manuscript within 60 days. If you anticipate any delay in its return, we ask that you let us know the expected resubmission date by replying to this email. Revised manuscripts received beyond 60 days may require evaluation and peer review similar to that applied to newly submitted manuscripts.

[LINK]

Sincerely,

William A. Petri, Jr.

Associate Editor

PLOS Pathogens

Vern Carruthers

Section Editor

PLOS Pathogens

Kasturi Haldar

Editor-in-Chief

PLOS Pathogens

orcid.org/0000-0001-5065-158X

Grant McFadden

Editor-in-Chief

PLOS Pathogens

orcid.org/0000-0002-2556-3526

Reviewer's Responses to Questions

**Part I - Summary**

Reviewer #1: Cernikova et al have addressed a longstanding question about the identity of PIP binding proteins in Giardia and their role in regulating endocytic trafficking. They used sensitive bioinformatics tools to identify candidate PIP binding proteins, modeled these proteins to test whether they had conserved residues involved in PIP and lipid binding, and then tested lipid binding activity with in vitro assays. They characterized the localization of these proteins and their relationship to clathrin and fluid phase endocytic cargo using STED super-resolution imaging. Further they explored the function of these proteins in Giardia using a combination of approaches including saturation of binding sites with over expressed PIP binding proteins, dominant negative mutants and chemical inhibitors. They performed crosslinking and pulldown experiments to identify the interactome of several validated lipid binding proteins and established that many of these proteins are in the same clathrin interaction network. They also performed EM demonstrating that some of the dominant negative proteins caused defects in PV morphology. Overall the paper is a major advance and represents a substantial amount of work. What detracts from the work is a focused concise message and a model for how these perturbations block endocytosis. In my opinion there is simply too much data packed into the main figures which will make tracking key points a challenge for the general readership of Plos Pathogens. While the manuscript initially focuses on PIP binding proteins the proteomics results lead to tangents that I think could be reserved for future studies. Rather than following up on new proteins pairwise co-localization studies of the identified lipid binding proteins would lead to a more comprehensive understanding of how these proteins are organized in relationship to each other.

Reviewer #2: Cernikova et al have studied the importance of PIPs in Giardia endocytosis. It is overall a well-performed study but some of studies have been performed earlier and the outline of the paper can be changed to make it more accessible.

**Part II – Major Issues: Key Experiments Required for Acceptance**

Reviewer #1: Major Issues:

1. Figure 2 The presented gradients blots seem inconsistent with the strip blots found in the supplement. For example PXD3 very clearly has a dominant preference for PI(3)P on the strip blot but this is less clear on the gradient blot. The number of replicates is not indicated

2. Another issue is that the methods indicates that the MBP fusions were lyophilized. Some proteins cannot tolerate lyophilization as this treatment could impact folding and activity. So were the proteins tested for activity before lyophilization? Of particular interest are the proteins that showed no binding activity. If activity was only tested after lyophilization, then the authors should repeat the lipid binding assays with proteins tested immediately after purification.

3. The authors indicate that their structure prediction identified key preserved residues. If they would mutate some of these for example in the most divergent of the identified proteins with lipid binding activity, this would help support the accuracy of the structure prediction. This could be performed in strip blots or through localization as one would predict the proteins of interest would no longer localize to PVs or bind the blotted lipids.

4. Figure 3. 2mM neomycin seems like a lot of antibiotic and the duration of treatment was not specified. I understand Giardia should not be sensitive to this antibiotic, but I wonder if the authors checked viability of the cells before the endocytosis assay. Also there are two publications that performed bioinformatics to assess phosphoinositide kinase signaling in Giardia and showed that LY29004, an inhibitor of phosphoinositide kinases, impacts cell growth (Cox 2006 and Hernandez 2007). Rather than trying to saturate PIP sites their existence can be blocked using wortmanin or LY294002. One would predict that this would result in altered protein localization and ESV morphology. Additionally, the authors measured the cell width then did a t-test on the median cell size. They should either do a Mann-Whitney U on the median or they should use a t-test to compare means. (Mann-Whitney U is probably better.)

5. Despite extensive super-resolution and EM imaging the authors do not propose any explanation for how their manipulations block endocytosis. Is it that the PVs are no longer open to uptake dextran or is it that they no longer can close so the dextran diffuses away during fixation?

6. Figure 7. How do the authors know that the NT-FYVE is generating a new compartment versus just aggregating? Also I don’t know what relevance this has to the biology of Giardia it seems like an artifact.

7. I have concerns about using the CWP1 promoter to drive over expression. Encystation is promoted by changes in cholesterol and pH which is accompanied by morphological changes that could impact PV function. The authors should, perhaps for their favorite protein, test whether they get similar phenotypes with the tet promoter system. Also it is surprising that the authors did not attempt to use any reverse genetics tools to assess the function of these proteins. Dominant negative and over expression studies can indicate a role but they can also produce misleading results whereas reverse genetics will cleanly indicate if a protein is required for a process of interest.

Reviewer #2: (No Response)

**Part III – Minor Issues: Editorial and Data Presentation Modifications**

Reviewer #1: Minor Issues:

1. Abstract, Author summary and Intro should start by saying “In model eukaryotes” to make it clear that the biology of Giardia is not being discussed.

2. The authors indicate that they could not identify C2 domain containing proteins yet a GiardiaDB text search of C2 yields four proteins. Some of which are phosphoinositide kinases which would be expected to bind PIPs? See GL50803_16728, GL50803_114201, GL50803_17406, GL50803_14855.

3. Figure 1 Panel B can be moved to the supplement as the relevant proteins are shown again in C-I. This also raises the point that the legend stops at C. For panel A, the current size of the DIC insets are too small to see anything, these could be expanded if the panel is moved into the supplement.

4. Several proteomics tables include cytoplasmic HSP70 but the text indicates that no clathrin uncoating proteins were found. Is cytoplasmic HSP70 not involved in clathrin uncoating?

5. Figure 5. While the fact that snare proteins are part of the PXD2 interactome is intriguing, I think including the localization in the main text detracts from a focused story. I recommend removal of panel B. Additionally I think all the interactome data should be combined into one figure to reduce the total number of figures.

6. I am unaware of any particular genetics nomenclature for Giardia, but the authors use of :: to indicate protein fusions is confusing as it is more commonly used to indicate promoter fusions. In particular I was confused by pEHA::GlPXD2 it took me a long time to figure out this is the endogenous promoter driving an N-terminal HA fusion to PXD2. Perhaps pE::HA-GlPXD2 or pE::HA:GlPXD2 would be easier to understand.

7. Figure 6. I think the identification of another dynamin homolog in Giardia is fascinating, but I think its inclusion distracts from a focused story. especially since there is no discussion of how mechanistically the mutant blocks endocytosis. I think a detailed study of this new dynamin and the more conventional dynamin is totally warranted and could be packaged as a focused study with EM work to examine how these proteins are preventing endocytosis. On the other hand if the authors were to perform EM to examine whether the mutants alter ESV morphology this could help explain the endocytosis defect.

8. For the proteomics analysis the methods do not indicate whether individual protein bands, entire lanes, or a certain number of segments were cut out for mass spec analysis?

9. Figure 8 legend. It isn’t clear which promoter was used to drive expression. The authors seem to use ectopic and overexpression interchangeably. I would consider ectopic to be on a plasmid not integrated into the genome and over expression to be when they use the CWP1 promoter to purposely drive expression beyond normal levels. The work would be easier to follow if this was clarified. Is 8D actually using the endogenous promoter as indicated in the graph?

10. Figure 9F dorsal view appears to be in a different orientation than the ventral view?

11. Figure 10. It would be helpful to include a key for the cartoon. Also the authors switched colors for dextran and HA in the middle of the paper. I recommend they always false color HA red and dextran green to match the model regardless of which fluorophores they used. The labels can simply say HA and Dextran and it will help the reader follow the work. 10D-G. In the interest of streamlining the paper I feel that one simplified overall interactome in the working model would be sufficient. The others could be in supplement.

Reviewer #2: 1. Line 44, 55 proteins in what organism(s). In all eukaryotes?

2. Lines 94 and 95. There are of course also amino acid, sugar and nucleoside transporters in the membrane that are crucial for nutrient uptake.

3. I lack more references to already published studies of Giardia PIP binding proteins in the introduction.

4. Line 124. In silico annotation techniques, what specifically is this?

5. Line 132. Not all PIP interacting proteins are associated with endocytosis.

6. Line 175. It is good to make it clear that the localizations were done in this study.

7. Line 192. It is not clear to me why these particular proteins were selected for further studies.

8. Line 197. Was overexpression with another promoter tested on the PXD5 protein to see a stronger signal? I see a signal in 1B.

9. Line 199. G. muris data is now available at NCBI, are all the proteins conserved there?

10. Line 210. This has already been done in an earlier study by Jana et al, 2017.

11. Line 230. Compare new data with old data from Jana. Add the PIP binding data to Table 1.

12. Line 245. How do the authors now that it is saturation, was a titration curve done?

13. Line 251. The controls could have been a parasite overexpressing another PIP domain containing protein not associated with the PVs.

14. Line 264, Neomycin kills Giardia, can this affect uptake?

15. Line 286. The whole interactome part to line 423 can be condensed and the results presented in one table. The follow up data specific for each protein can complement this

PLOS authors have the option to publish the peer review history of their article (what does this mean?). If published, this will include your full peer review and any attached files.

Reviewer #1: No

Reviewer #2: No

---

## [Editor Report · Decision Letter 1]

14 Jan 2020

Dear Prof. Hehl,

We are pleased to inform that your manuscript, "Phosphoinositide-binding proteins mark, shape and functionally modulate highly-diverged endocytic compartments in the parasitic protist Giardia lamblia", has been editorially accepted for publication at PLOS Pathogens. 

Before your manuscript can be formally accepted and sent to production, you will need to complete our formatting changes, which you will receive by email within a week. Please note that your manuscript will not be scheduled for publication until you have made the required changes.

IMPORTANT NOTES

(1) Please note, once your paper is accepted, an uncorrected proof of your manuscript will be published online ahead of the final version, unless you’ve already opted out via the online submission form. If, for any reason, you do not want an earlier version of your manuscript published online or are unsure if you have already indicated as such, please let the journal staff know immediately at plospathogens@plos.org.

(2) Copyediting and Proofreading: The corresponding author will receive a typeset proof for review, to ensure errors have not been introduced during production. Please review the PDF proof of your manuscript carefully, as this is the last chance to correct any errors. Please note that major changes, or those which affect the scientific understanding of the work, will likely cause delays to the publication date of your manuscript. 

(3) Appropriate Figure Files: Please remove all name and figure # text from your figure files. Please also take this time to check that your figures are of high resolution, which will improve the readbility of your figures and help expedite your manuscript's publication. Please note that figures must have been originally created at 300dpi or higher. Do not manually increase the resolution of your files. For instructions on how to properly obtain high quality images, please review our Figure Guidelines, with examples at: http://journals.plos.org/plospathogens/s/figures.

(4) Striking Image: Please upload a striking still image to accompany your article if one is available (you can include a new image or an existing one from within your manuscript). Should your paper be accepted, this image will be considered for our monthly issue image and may also appear on our website to feature your article. Please upload this as a separate file, selecting "striking image" as the file type upon upload. Please also include a separate "Other" file with a caption, including credits and any potential copyright information. Please do not include the caption in the main article file. If your image is from someone other than yourself, please ensure that the artist has read and agreed to the terms and conditions of the Creative Commons Attribution License at http://journals.plos.org/plospathogens/s/content-license. Please note that PLOS cannot publish copyrighted images.

(5) Press Release or Related Media: If your institution or institutions have a press office, please notify them about your upcoming paper at this point, to enable them to help maximize its impact. If they will be preparing press materials for this manuscript, please inform our press team in advance at plospathogens@plos.org as soon as possible. We ask that you contact us within one week to plan ahead of our fast Production schedule. If you need to know your paper's publication date for related media purposes, you must coordinate with our press team, and your manuscript will remain under a strict press embargo until the publication date and time. This means an early version of your manuscript will not be published ahead of your final version. 

(6)  PLOS requires an ORCID iD for all corresponding authors on papers submitted after December 6th, 2016. Please ensure that you have an ORCID iD and that it is validated in Editorial Manager.  To do this, go to ‘Update my Information’ (in the upper left-hand corner of the main menu), and click on the Fetch/Validate link next to the ORCID field.  This will take you to the ORCID site and allow you to create a new iD or authenticate a pre-existing iD in Editorial Manager

(7) Update your Profile Information: Now that your manuscript has been provisionally accepted, please log into Editorial Manager and update your profile, if needed. Go to https://www.editorialmanager.com/ppathogens, log in, and click on the "Update My Information" link at the top of the page. Please update your user information to ensure an efficient production and billing process. 

(8) LaTeX users only: Our staff will ask you to upload a TEX file in addition to the PDF before the paper can be sent to typesetting, so please carefully review our Latex Guidelines http://journals.plos.org/plospathogens/s/latex in the meantime.

(9) If you have associated protocols in protocols.io, please ensure that you make them public before publication to guarantee immediate access to the methodological details.

Best regards,

William A. Petri, Jr.

Associate Editor

PLOS Pathogens

Vern Carruthers

Section Editor

PLOS Pathogens

Kasturi Haldar

Editor-in-Chief

PLOS Pathogens

orcid.org/0000-0001-5065-158X

Michael Malim

Editor-in-Chief

PLOS Pathogens

orcid.org/0000-0002-7699-2064
---

## [Editor Report · Acceptance letter]

5 Feb 2020

Dear Prof. Hehl,

We are delighted to inform you that your manuscript, "Phosphoinositide-binding proteins mark, shape and functionally modulate highly-diverged endocytic compartments in the parasitic protist *Giardia lamblia*," has been formally accepted for publication in PLOS Pathogens.

Best regards,

Kasturi Haldar

Editor-in-Chief

PLOS Pathogens

orcid.org/0000-0001-5065-158X

Michael Malim

Editor-in-Chief

PLOS Pathogens

orcid.org/0000-0002-7699-2064